# LEARNING A LINEAR DELAY SURROGATE MODEL FOR TIMING-DRIVEN CHIP GLOBAL PLACEMENT

## ABSTRACT

Timing-driven global placement (GP) is a critical step in chip physical design, where the objective is to determine the physical locations of millions of cells to optimize signal delays and satisfy timing constraints. Existing GP algorithms commonly rely on gradient-based optimization, which requires the placement objective to be differentiable with respect to cell coordinates. However, timing evaluation—particularly the delay computation—is inherently complex and typically non-differentiable, making it difficult to integrate into gradient-based GP algorithms. To address this challenge, we propose **LiTPlace**, a **L**earning-based **T**iming-driven global placement framework, which learns a differentiable surrogate model to predict signal delays for timing-aware gradient-based optimization. To the best of our knowledge, the application of machine learning (ML) in timing-driven GP remains underexplored in previous works. At the core of LiTPlace is a graph neural network (GNN) inspired by the signal propagation in chip circuits, which predicts signal delays based on the netlist graph structure and the placement geometry. To ensure compatibility with gradient-based optimization, we design the GNN architecture so that its output is approximately a linear function of a set of geometric distance statistics, enabling efficient and stable gradient computation with respect to cell coordinates. Experiments on 28 chip designs from widely used benchmarks demonstrate that LiTPlace significantly improves timing quality, achieving an average improvement of $19.2\%$ in TNS and $7.7\%$ in WNS, which are two key metrics to quantify the chip timing quality.

## 1    INTRODUCTION

Electronic Design Automation (EDA) tools are crucial in modern chip design, enabling designers to manage the growing complexity and scale of integrated circuits (MacMillen et al., 2000; Markov et al., 2012). A central goal across the EDA workflow is to optimize physical and performance metrics, among which **timing performance** is especially critical, as it determines the maximum operating frequency and reflects whether signals propagate reliably within required timing constraints (Rabaey et al., 2002; Wang et al., 2009). While multiple design stages—such as logic synthesis, clock tree synthesis (CTS), and routing—affect timing, the **placement** stage plays a particularly pivotal role. It determines the physical locations of millions of chip components—including standard cells and macros—which directly affect the signals paths and propagation delays. Suboptimal placement can lead to late-arriving signals that violate timing constraints, thus causing functional failures. Therefore, **timing-driven placement** is a fundamental task in the EDA workflow (Wang et al., 2024b; Xue et al., 2025; Geng et al., 2025). Placement is typically divided into three stages: **macro placement (MP)**, **global placement (GP)**, and **detailed placement (DP)**. MP arranges large functional blocks called macros; GP determines approximate locations for a large amount of standard cells; and DP fine-tunes these locations to meet strict design rules. Among these, GP is the first stage to perform full-chip placement and has the most substantial influence on the overall timing performance (Cheng et al., 2018; Shi et al., 2025; Fu et al., 2024).

Despite its importance, **timing-driven global placement (GP)** remains challenging due to its large scale, continuous search space, and the difficulty of integrating accurate timing evaluation into the optimization loop. As shown in Figure 1, GP involves positioning millions of standard cells on a chip layout, resulting in an extremely enormous and continuous design space. To handle this, existing GP algorithms primarily rely on **gradient-based optimization**, which iteratively updates cell coordinates

Figure 1: Comparison between macro placement (MP) and global placement (GP).

based on gradients of a differentiable placement objective (Lin et al., 2019; 2020; Gu et al., 2020; Liao et al., 2022). However, incorporating timing evaluation into such optimization is nontrivial, as timing analysis—especially delay computation—is computationally expensive and does not yield gradients with respect to cell coordinates. As a result, many approaches rely on differentiable, yet heuristic surrogates, such as approximated half-perimeter wirelength (HPWL), which are empirically correlated with timing performance (Lin et al., 2019; Guo & Lin, 2022). Such gap between heuristics and actual objectives can lead to suboptimal results, highlighting an opportunity for machine learning (ML) methods to offer more accurate and differentiable surrogate models.

Recently, ML techniques have shown great success across various design stages (Chen et al., 2024), including RTL code generation and logic synthesis (Thakur et al., 2024; Wang et al., 2024a;c; Lai et al., 2025). In particular, ML-based approaches—such as reinforcement learning (RL) and black-box optimization (BBO)—have achieved promising results in MP by replacing hand-crafted heuristics with data-driven policies (Mirhoseini et al., 2021; Lai et al., 2022; 2023; Cheng & Yan, 2021; Cheng et al., 2022; Shi et al., 2023; Geng et al., 2025). However, to the best of our knowledge, the application of ML to GP remains largely underexplored, owing to the combined challenges of large scale and structural complexity.

In this paper, we propose **LiTPlace**, a **L**earning-based **T**iming-driven global placement framework. The key idea of LiTPlace is to learn a signal delay predictor that is **differentiable** with respect to cell coordinates, thus enabling timing objectives to be directly integrated into gradient-based optimization. At the core of LiTPlace is a propagation-based graph neural network (GNN) that predicts delays based on the netlist structure and placement geometry. The GNN is inspired by the way timing signals propagate through the digital circuits, so as to model delay dependencies accurately. Importantly, to ensure compatibility with gradient-based optimization, we design the GNN architecture such that its output is approximately a **linear function** of pairwise cell distance statistics[1], enabling efficient and stable gradient computation with respect to cell coordinates. We evaluate LiTPlace on two widely used benchmark suites containing 28 chip designs from diverse domains. Experiments demonstrate that LiTPlace significantly improves placement timing quality, achieving an average improvement of 19.2% in total negative slack (TNS) and 7.7% in worst negative slack (WNS), which are two key metrics to assess the chip timing performance.

## 2 PRELIMINARIES

We begin by introducing some necessary background to help readers understand our task. Supplemental background is in Appendix A, and a discussion of related work is in Appendix B.

### 2.1 BASICS OF CHIP DESIGN

A digital chip mainly consists of two types of components: **cells** and **macros**. **Cells** are the basic building blocks of the chip circuit. Cells include *registers*, which store signals, and *logic gates*, which perform logical operations on signals. **Macros** are large, pre-designed modules made up of many cells, and they typically implement more complex logic relationships. Each component contains a set of **pins**, which serve as connection points for signal transmission. Pins are categorized into *input pins*, which receive signals from upstream components, and *output pins*, which send signals to downstream components. **Nets** represent connections of pins across components, allowing signals to propagate across the circuit. Each net includes one output pin (called *driver pin*) to send the signal onto the net, and one or more input pins (called *load pins*) that receive the signal. Figure 2(a) illustrates these basic concepts. When the chip operates, signals are launched from input pins, propagate through a

---

[1]Here, "linear" is an approximate description to aid understanding. Precise formulation is in Theorem 1.

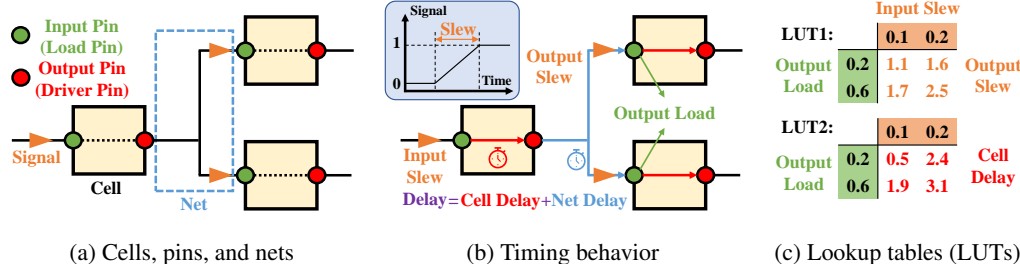

(a) Cells, pins, and nets      (b) Timing behavior      (c) Lookup tables (LUTs)

Figure 2: Illustrations of basic concepts in timing-driven global placement.

chain of cells and nets, and eventually arrive at output pins. To ensure correct functionality, these signals must arrive at their destinations within specified time constraints. This leads to the concept of **timing performance**, which measures how well a chip ensures that signals arrive on time.

Key concepts related to timing performance include: **delay**, **arrival time**, **slack**, **worst negative slack (WNS)**, and **total negative slack (TNS)**, which are illustrated in Figure 2(b). **Delay** refers to the time a signal takes to travel from one pin to another. It can be categorized into *cell delay*—the delay from an input pin to an output pin within the same cell—and *net delay*—the delay from the output pin of one cell to the input pin of another cell via a connecting net. **Arrival time** is the time a signal takes to reach its destination. It is computed by accumulating delays along its propagation path. **Slack** refers to the difference between the required and the actual arrival time. A negative slack indicates a timing violation. Two standard metrics derived from slack are commonly used to evaluate chip timing performance: **worst negative slack (WNS)** that pinpoints the most severe timing violation in the design, and **total negative slack (TNS)** that sums all negative slacks across the circuit. Smaller magnitudes of WNS and TNS (i.e., values closer to zero) indicate better timing performance.

## 2.2 STATIC TIMING ANALYSIS

**Static Timing Analysis (STA)** is a standard method to estimate the timing behavior of digital circuits (Bhasker & Chadha, 2009). It typically proceeds in three steps: (1) computing delays, (2) calculating signal arrival times based on delays, and (3) verifying whether these arrival times meet required timing constraints.

Computing the delay is a critical yet timing-consuming step in STA, which relies on a timing model provided by a `.lib` file. This file, as a key part of the technology library, is supplied by the technology provider (i.e., the foundry) and specifies the functional and timing characteristics of each standard cell. In post-placement STA, **net delay** is relatively straightforward to estimate. It primarily depends on the physical length and topology of the net, as well as the total **capacitance** of the **load pins**. These pin capacitance values are provided in the `.lib` file. In contrast, the **cell delay** is more complex to compute. It depends on two key factors: the **input slew** and the **output load**. The **slew** describes how quickly a signal transitions from low to high (or vice versa). The **output load** is the total capacitance seen at the output pin, determined by the layout of the connecting net and the downstream load pins. Each cell receives an input slew and generates an output slew, which is affected by the output load imposed by the connected net and its load pins.

Given the aforementioned two inputs, the STA engine uses pre-characterized Look-Up Tables (LUTs) in the `.lib` file to obtain the corresponding output slew and cell delay, as illustrated in Figure 2(c). The values are obtained via bilinear interpolation from discrete LUT entries. The output slew then propagates forward as the input slew to the next cell. This forward-propagation process continues across the circuit, allowing STA to recursively compute delays throughout the netlist.

## 2.3 GLOBAL PLACEMENT

**Global placement (GP)** is a core stage in chip physical design. In this stage, the positions of macros are fixed, and the task is to determine the locations of standard cells on the chip canvas, subject to constraints such as non-overlap, while optimizing the overall timing performance.

GP is typically formulated as a continuous optimization problem and then solved by gradient-based methods (Lin et al., 2019; 2020; Gu et al., 2020; Liao et al., 2022). The objective is reformulated as to minimize the overall wirelength, based on the heuristic that shorter interconnects generally lead to lower sinal delays and improved timing performance. Since routing has not yet been performed at this stage, the actual wirelength is unavailable. Instead, a widely adopted surrogate is the **half-perimeter wirelength (HPWL)**, defined as the sum of the horizontal and vertical spans of the bounding box enclosing all pins of a net. HPWL is favored for its simplicity and empirical effectiveness in approximating wirelength. The global placement problem is then often relaxed into the following optimization problem:

$$\min_{(\boldsymbol{x}, \boldsymbol{y})} \sum_{\text{net} \in \mathcal{N}} \tilde{W}_{\text{net}}(\boldsymbol{x}, \boldsymbol{y}) + \lambda \cdot D(\boldsymbol{x}, \boldsymbol{y}), \tag{1}$$

where $(\boldsymbol{x}, \boldsymbol{y})$ denotes the coordinates of all cells, $\mathcal{N}$ is the set of nets, $\tilde{W}_{\text{net}}$ represents a smoothed approximation of HPWL for each net, $D$ is a density penalty term that discourages overlap, and $\lambda$ is a hyperparameter. Both terms are designed to be differentiable with respect to the coordinates $\boldsymbol{x}$ and $\boldsymbol{y}$, allowing the use of gradient-based methods to directly optimize $\boldsymbol{x}$ and $\boldsymbol{y}$.

## 3 MOTIVATION

To incorporate timing optimization into global placement, a natural idea is to replace the expensive and non-differentiable delay estimation in STA with a lightweight, differentiable surrogate model. Specifically, we seek to learn a predictor $f_{\boldsymbol{\theta}}(\mathcal{G}; \boldsymbol{x}, \boldsymbol{y})$ that estimates signal delays based on the netlist $\mathcal{G}$ and cell coordinates $(\boldsymbol{x}, \boldsymbol{y})$, while remaining differentiable with respect to $\boldsymbol{x}$ and $\boldsymbol{y}$. This allows the surrogate to be directly integrated into gradient-based placement optimization. However, using raw coordinates $\boldsymbol{x}$ and $\boldsymbol{y}$ as model inputs introduces unnecessary complexity and sensitivity to global shifts. A key observation is that the delays are primarily governed by the **relative positions** of pins, specifically, the pairwise distances between pins. Motivated by this, we reformulate the delay predictor to take as input the pairwise distances $\boldsymbol{d}$ between cells, yielding a model $f_{\boldsymbol{\theta}}(\mathcal{G}; \boldsymbol{d})$ that captures the most relevant geometric features while preserving differentiability. Yet, two technical questions remain . (1) How can we effectively model delay dependencies across the netlist to achieve accurate delay prediction? (2) How can we design the predictor architecture to support efficient gradient computation with respect to the pairwise distances $\boldsymbol{d}$?

**(1) Capturing Delay Dependencies via Propagation-Based Graph Neural Network**  As discussed in Section 2.2, STA performs delay computation through a forward propagation process, where the delay at each pin depends on the slews of its upstream nodes. To model this dependency, we design a propagation-based graph neural network (GNN), denoted as $\text{GNN}_{\boldsymbol{\theta}}(\mathcal{G}; \boldsymbol{d})$, which simulates the signal timing propagation along the netlist. The GNN processes the circuit graph in topological order, passing messages from upstream to downstream nodes. By aligning the message-passing dynamics with the actual signal flow, this architecture accurately models timing dependencies across the circuit.

**(2) Enabling Efficient Gradient Computation via Linear Propagation**  While the propagation-based GNN can effectively model delay behavior, integrating it into a gradient-based placement framework introduces new challenges. In general, GNNs are structurally deep and complex. Computing the gradients $\nabla_{\boldsymbol{d}} \text{GNN}_{\boldsymbol{\theta}}(\mathcal{G}; \boldsymbol{d})$ typically requires full backpropagation through multiple message-passing layers, which can be both computationally expensive and potentially unstable during optimization. To address this, we design the GNN architecture such that the predicted delays are, *loosely speaking*, **linear** with respect to the input distances $\boldsymbol{d}$. This linearity is preserved even through multiple message-passing layers, as **the composition of linear operations remains linear**[2]. As a result, the gradients $\nabla_{\boldsymbol{d}} \text{GNN}_{\boldsymbol{\theta}}(\mathcal{G}; \boldsymbol{d})$ can be computed analytically and efficiently, enabling efficient and stable integration into the gradient-based placement frameworks.

## 4 METHODOLOGY

This section presents our proposed approach, LiTPlace. An overview of LiTPlace is illustrated in Figure 3. In Section 4.1, we introduce the graph representation of the circuit netlist. Then,

---

[2]We refer to Theorem 1 for the formal definition of such approximate linearity.

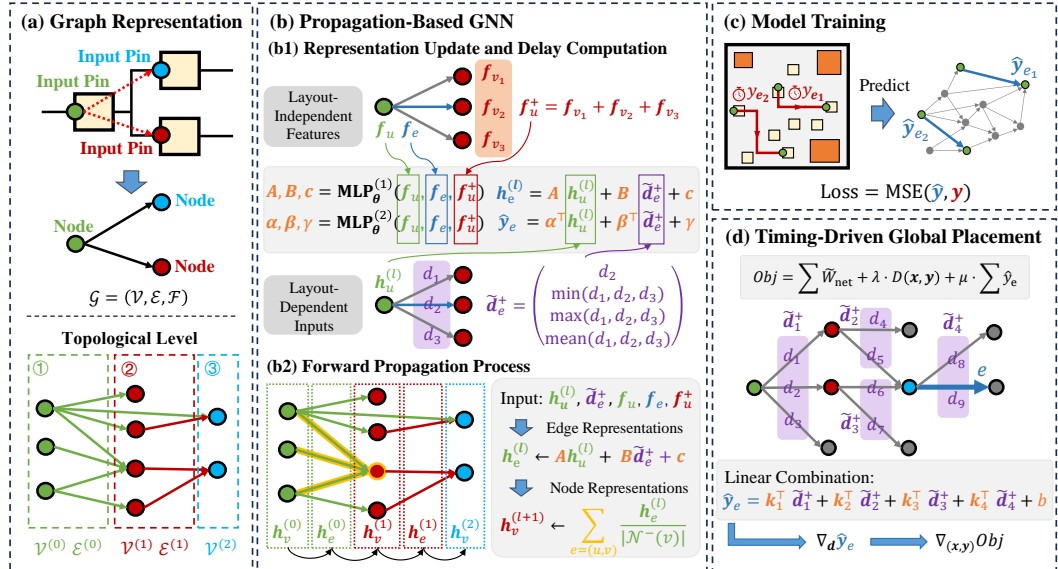

Figure 3: **Overview of LiTPlace.** (a) We represent the circuit netlist as as a DAG $\mathcal{G} = (\mathcal{V}, \mathcal{E}, \mathcal{F})$, which is partitioned into topological levels reflecting the signal propagation order. (b) We use a propagation-based GNN to predict edge delays. At each level, we use layout-independent features to generate coefficients for a linear function that combines the previous node representations and geometric distance statistics to produce the edge representations. The propagation process iterates by alternately updating edge and node representations in a level-wise manner. (c) We train the GNN model to predict edge delays in a supervised manner. (d) During timing-driven global placement, the predicted delays are incorporated into the optimization objective. Thanks to the model's linear structure, gradients with respect to cell coordinates can be computed efficiently.

Section 4.2 details the architecture of our delay prediction model. Finally, in Section 2.3, we describe how the learned delay predictor is integrated into gradient-based placement optimization. More implementation details can be found in Appendix C.

## 4.1 GRAPH REPRESENTATION AND TOPOLOGICAL LEVEL

As shown in Figure 3(a), we represent the circuit netlist as a directed acyclic graph (DAG), denoted as $\mathcal{G} = (\mathcal{V}, \mathcal{E}, \mathcal{F})$, where each node $v \in \mathcal{V}$ corresponds to an input pin, and each edge $e \in \mathcal{E}$ represents a signal connection between two pins. An edge $e = (u, v)$ is added if a signal is propagated from an input pin $u$, through a cell and its connecting net, to the input pin $v$ of a downstream cell. Each edge is then associated a delay, i.e., the sum of a cell delay (the delay from an input pin to the output pin within the cell) and a net delay (the delay from the output pin to the next cell's input pin via the net). We refer to this combined delay as an edge delay. Our objective is to predict the edge delay $y_e$ for each edge $e \in \mathcal{E}$, formulated as a supervised edge-level regression problem. For a given layout, each edge $e = (u, v)$ is also associated with a distance $d_e = d(u, v)$, i.e., the distance between the two associated input pins in the layout. We denote the concatenation of all such distances as $\boldsymbol{d}$.

We annotate the graph with timing-relevant features $\mathcal{F}$ on both nodes and edges. Each node is assigned attributes including the pin capacitance, in-degree, and out-degree within the DAG. Each edge is assigned features derived from the standard cell it passes through. Specifically, we extract the corresponding Look-Up Tables (LUTs) from the `.lib` file, and apply principal component analysis (PCA) to embed them into low-dimensional feature vectors.

We then introduce some structural notations over the DAG. The **topological level** of a node or an edge is defined as the maximum number of hops from any source node (i.e., a node with zero in-degree). We use $\mathcal{V}^{(l)}$ and $\mathcal{E}^{(l)}$ to denote the sets of nodes and edges at topological level $l$, respectively. For each node $u \in \mathcal{V}$, we define its predecessor and successor neighborhoods as $\mathcal{N}^-(u) = \{v \mid (v, u) \in E\}$ and $\mathcal{N}^+(u) = \{v \mid (u, v) \in E\}$, respectively. Additional details can be found in Appendix C.1.

## 4.2 PROPAGATION-BASED GNN FOR EDGE DELAY PREDICTION

Based on the graph representation introduced in Section 4.1, we can employ a propagation-based graph neural network (GNN) to predict the edge delay $y_e$ for each edge $e \in \mathcal{E}$. The GNN takes two types of inputs: (1) the netlist structure $\mathcal{G}$, which is layout-independent, and (2) the pairwise pin distances $\boldsymbol{d}$ for given layouts. The prediction process is then formulated as:

$$\hat{\boldsymbol{y}} = \text{GNN}_{\boldsymbol{\theta}}(\mathcal{G}; \boldsymbol{d}), \tag{2}$$

where $\hat{\boldsymbol{y}} \in \mathbb{R}^{|\mathcal{E}|}$ denotes the predicted edge delays.

Our GNN architecture is illustrated in Figure 3(b). It is inspired by the forward propagation mechanism in STA. Following the topological level order, the GNN performs message passing from source nodes to leaf nodes, and sequentially compute the node and edge representations. At each level $l \in \{0, 1, \ldots, L\}$, we denote the node representation as $\boldsymbol{h}_u^{(l)} \in \mathbb{R}^k$ for each $u \in \mathcal{V}^{(l)}$, and the edge representation as $\boldsymbol{h}_e^{(l)} \in \mathbb{R}^k$ for each $e \in \mathcal{E}^{(l)}$, where $k$ is the embedding dimension. At the initial level $l = 0$, all node representations are initialized as zero vectors, i.e., $\boldsymbol{h}_u^{(0)} = \boldsymbol{0}$. At level $l$, we assume that node representations $\boldsymbol{h}_u^{(l)}$ have been computed, and we now describe how to construct edge representations $\boldsymbol{h}_e^{(l)}$ for $e = (u, v) \in \mathcal{E}^{(l)}$, and node representations $\boldsymbol{h}_v^{(l+1)}$ for the next level.

To enable accurate delay prediction, the edge representations are designed to encode timing-relevant information derived from both the netlist structure and physical layout. As introduced in Section 2.2, the delays depend on multiple factors, including input slew, pin capacitance, output load, layout geometry, and the cell timing models defined by Look-Up Tables (LUTs). To model these dependencies, the representation of each edge $e = (u, v) \in \mathcal{E}^{(l)}$ is computed from the following five components:

(1) **Source node representation** $\boldsymbol{h}_u^{(l)}$. It encodes propagated timing signal information from upstream nodes, such as input slew. This information is essential for estimating the cell delay.

(2) **Pooled distance statistics** $\tilde{\boldsymbol{d}}_e^+$. Notably, delay is influenced not only by the distance $d_e = d(u, v)$, but also by all successors of node $u$, which contribute to the output load and thus affect both cell and net delays. Therefore, we define the set of distances $\boldsymbol{d}_u^+ = \{d(u, v') : v' \in \mathcal{N}^+(u)\}$ to capture the placement local geometry, where $d(u, v')$ denotes the physical distance between pins $u$ and $v'$. To obtain a fixed-size representation, we apply a four-dimensional pooling operator:

$$\tilde{\boldsymbol{d}}_e^+ = \left( d_e, \min(\boldsymbol{d}_u^+), \max(\boldsymbol{d}_u^+), \text{mean}(\boldsymbol{d}_u^+) \right)^\top. \tag{3}$$

(3) **Source node feature vector** $\boldsymbol{f}_u$. It includes pin-level attributes of node $u$, such as capacitance, in-degree, and out-degree, which are relevant for estimating cell behavior.

(4) **Edge feature vector** $\boldsymbol{f}_e$. It contains timing-related features of edge $e$, specifically the PCA-compressed LUT embeddings used to model the cell delay and output slew.

(5) **Pooled successor node feature vector** $\boldsymbol{f}_u^+$. We define $\mathcal{F}_u^+ = \{\boldsymbol{f}_{v'} : v' \in N^+(u)\}$, which includes the features of all successor nodes and affects both cell and net delays. We aggregate these features via a sum pooling to produce a fixed-size representation:

$$\boldsymbol{f}_u^+ = \sum_{\boldsymbol{f}_{v'} \in \mathcal{F}_u^+} \boldsymbol{f}_{v'} \tag{4}$$

We then design a mapping function $\phi_{\boldsymbol{\theta}}$ that combines these five inputs to obtain edge representations:

$$\boldsymbol{h}_e^{(l)} = \phi_{\boldsymbol{\theta}} \left( \boldsymbol{h}_u^{(l)}, \tilde{\boldsymbol{d}}_e^+; \boldsymbol{f}_u, \boldsymbol{f}_e, \boldsymbol{f}_u^+ \right). \tag{5}$$

Notably, among these, the first two inputs $\boldsymbol{h}_u^{(l)}$ and $\tilde{\boldsymbol{d}}_u^+$ are **layout-dependent**, i.e., they depend on the edge distances $\boldsymbol{d}$. The remaining three features, $\boldsymbol{f}_u$, $\boldsymbol{f}_e$, and $\boldsymbol{f}_u^+$, are **layout-independent**, as they are determined solely by the netlist itself, but not the layout. **Our key technical insight** is to ensure that $\phi_{\boldsymbol{\theta}}$ is **linear** with respect to the layout-dependent inputs, allowing the resulting representations to support efficient and stable gradient computation. Specifically, we compute:

$$\boldsymbol{h}_e^{(l)} = \boldsymbol{A}_e \boldsymbol{h}_u^{(l)} + \boldsymbol{B}_e \tilde{\boldsymbol{d}}_e^+ + \boldsymbol{c}_e, \tag{6}$$

where $\boldsymbol{A}_e \in \mathbb{R}^{k \times k}$, $\boldsymbol{B}_e \in \mathbb{R}^{k \times 4}$, and $\boldsymbol{c}_e \in \mathbb{R}^k$ are coefficients generated by a multilayer perceptrons (MLP) conditioned on the layout-independent features:

$$\boldsymbol{A}_e, \boldsymbol{B}_e, \boldsymbol{c}_e = \text{MLP}_{\boldsymbol{\theta}}^{(1)} \left( \boldsymbol{f}_u, \boldsymbol{f}_e, \boldsymbol{f}_u^+ \right). \tag{7}$$

Once all edge representations $\boldsymbol{h}_e^{(l)}$ are computed at level $l$, the node representations at level $l+1$ are obtained via mean aggregation over the incoming edges:

$$\boldsymbol{h}_v^{(l+1)} = \frac{1}{|\mathcal{N}^-(v)|} \sum_{e=(u,v)\in\mathcal{E}} \boldsymbol{h}_e^{(l)}. \tag{8}$$

To predict the final edge delay $\hat{y}_e$ for each $e = (u, v)$, we employ a decoder with a similar linear structure. The decoder takes as input the same set of features used to construct the edge representation:

$$\hat{y}_e = \boldsymbol{\alpha}_e^\top \boldsymbol{h}_u^{(l)} + \boldsymbol{\beta}_e^\top \tilde{\boldsymbol{d}}_e^+ + \gamma_e, \tag{9}$$

where $\boldsymbol{\alpha}_e \in \mathbb{R}^k$, $\boldsymbol{\beta}_e \in \mathbb{R}^4$, and $\gamma_e \in \mathbb{R}$ are coefficients generated by another MLP, also conditioned on layout-independent features:

$$\boldsymbol{\alpha}_e, \boldsymbol{\beta}_e, \gamma_e = \text{MLP}_{\boldsymbol{\theta}}^{(2)}(\boldsymbol{f}_u, \boldsymbol{f}_e, \boldsymbol{f}_u^+). \tag{10}$$

**Message-Passing Interpretation.** The above formulation aligns precisely with the standard message-passing paradigm of graph neural networks. In each propagation step, the edge representation $\boldsymbol{h}_e^{(l)}$ serves as the message sent from the source node $u$ to the target node $v$, which is defined as Equation 6. The subsequent node update corresponds to a classical AGGREGATE operation, followed by a COMBINE step that reduces to simply taking the aggregated message as the new node representation. In this way, the architecture preserves the full semantics of message passing—messages flow along edges, are aggregated at nodes, and are propagated iteratively through the graph.

**Preservation of Linearity in Propagation.** A central property of our model architecture is that, loosely speaking, the predicted delay $\hat{y}_e$ for each edge is a **linear function of layout-dependent inputs**, namely the pooled distance statistics. This structural linearity arises from our architectural design: both the edge representations $\boldsymbol{h}_e^{(l)}$ and the decoder outputs $\hat{y}_e$ are constructed as linear functions of layout-dependent variables, with coefficients entirely determined by layout-independent features. We formally state this property in the following theorem.

**Theorem 1.** *Given a circuit netlist $\mathcal{G} = (\mathcal{V}, \mathcal{E}, \mathcal{F})$, for any topological level $l \in \mathbb{N}$ and edge $e = (u, v) \in \mathcal{E}^{(l)}$, there exists a set of vectors $\left\{ \boldsymbol{a}_{e,e'} \in \mathbb{R}^4 : e' \in \bigcup_{i=0}^l \mathcal{E}^{(i)} \right\}$ and a scalar bias $b_e \in \mathbb{R}$, such that for any pairwise distance configuration $\boldsymbol{d}$, the predicted delay satisfies:*

$$\hat{y}_e(\boldsymbol{d}) = \sum_{e' \in \bigcup_{i=0}^l \mathcal{E}^{(i)}} \boldsymbol{a}_{e,e'}^\top \tilde{\boldsymbol{d}}_{e'}^+ + b_e, \tag{11}$$

*where $\tilde{\boldsymbol{d}}_{e'}^+ \in \mathbb{R}^4$ is the pooled distance statistic vector associated with edge $e'$ (see Equation (3)).*

Theorem 1 demonstrates the preservation of linearity throughout the propagation process. As a result, the gradients can be efficiently computed without backpropagating through the full GNN:

$$\nabla_{\boldsymbol{d}} \hat{y}_e(\boldsymbol{d}) = \sum_{e' \in \bigcup_{i=0}^l \mathcal{E}^{(i)}} \left( \nabla_{\boldsymbol{d}} \tilde{\boldsymbol{d}}_{e'}^+ \right)^\top \boldsymbol{a}_{e,e'}, \tag{12}$$

where the coefficients $\boldsymbol{a}_{e,e'}$ are layout-independent and thus can be pre-computed for any given netlist. Additional architectural details are in Appendix C.2, and the proof of Theorem 1 is in Appendix C.3.

**Summary of the propagation-based GNN**  While the propagation equations in our architecture are linear with respect to layout-dependent variables, the overall model is not globally linear. The key reason is that all linear coefficients—such as $\boldsymbol{A}_e$, $\boldsymbol{B}_e$, $\boldsymbol{c}_e$, $\boldsymbol{\alpha}_e$, $\boldsymbol{\beta}_e$, and $\gamma_e$—are generated by MLPs conditioned solely on layout-independent circuit features, including LUT-based timing characteristics, pin capacitance, and fan-out structure. These MLPs introduce essential nonlinearity that allows the model to capture complex circuit-level timing behaviors (e.g., slew–load interactions encoded in LUTs), which cannot be expressed by a single global linear transformation. At the same time, by enforcing linearity with respect to geometric distance statistics, the model guarantees stable and efficient gradient computation for timing-driven placement. This architectural separation—nonlinear modeling of circuit-intrinsic timing behavior and linear modeling of layout-induced geometric effects—is fundamental to achieving both predictive accuracy and differentiability.

### 4.3 TIMING-DRIVEN GLOBAL PLACEMENT WITH LINEAR DELAY SURROGATE

As illustrated in Figure 3(c), we train the delay surrogate to predict edge delays in a supervised manner. Training details are in Appendix C.4. Once trained, the learned model can predict delays $\hat{y}_e(\boldsymbol{d}) = \mathrm{GNN}_{\boldsymbol{\theta}}(\mathcal{G}; \boldsymbol{d}; e)$ for any given netlist $\mathcal{G}$ and layout geometry $\boldsymbol{d}$. To incorporate timing optimization into global placement, we augment the traditional objective in Equation (1) with a timing penalty term based on predicted delays, as illustrated in Figure 3(d). To better model the **critical paths**, which are the true timing bottlenecks of the design, we further design our objective to focus on the top-$K$ timing-critical paths. Specifically, after every fixed number of optimization steps, we extract a set $\mathcal{P}_K$ of $K$ critical paths $P$ with the highest cumulative predicted delays $\sum_{e \in \mathcal{E}(P)} \hat{y}_e(\boldsymbol{d})$. We restrict the timing penalty to only these paths, resulting in the following objective:

$$\min_{\boldsymbol{x}, \boldsymbol{y}} \sum_{\mathrm{net} \in \mathcal{N}} \tilde{W}_{\mathrm{net}}(\boldsymbol{x}, \boldsymbol{y}) + \lambda \cdot D(\boldsymbol{x}, \boldsymbol{y}) + \mu \cdot \sum_{P \in \mathcal{P}_K} \sum_{e \in \mathcal{E}(P)} \hat{y}_e(\boldsymbol{d}), \tag{13}$$

where $\tilde{W}_{\mathrm{net}}$ is the smoothed HPWL, $D$ is the density penalty, and $\lambda, \mu$ are hyperparameters.

Thanks to the linear structure of our GNN architecture, as illustrated in Theorem 1, each predicted delay $\hat{y}_e(\boldsymbol{d})$ can be viewed as a linear function (with a biased term) of distance statistics $\tilde{\boldsymbol{d}}_{e'}^+$. Consequently, the additional timing penalty is also a linear function, with coefficients determined solely by layout-independent features. We further provide an efficient procedure to pre-compute these coefficients in $\mathcal{O}(|\mathcal{E}|)$ time, as detailed in Algorithm 1, Appendix C.5. Once computed, the gradients can be efficiently obtained as shown in Equation (12), without backpropagating through the GNN. This enables efficient integration of timing objectives into gradient-based placement with negligible additional backpropagation cost. More implementation details can be found in Appendix C.5.

## 5 EXPERIMENTS

### 5.1 EXPERIMENTAL SETUP

**Benchmarks**  We evaluate LiTPlace on two benchmark suites: `ICCAD2015` and `ChiPBench`. `ICCAD2015` (Kim et al., 2015) originates from the timing-driven placement contest at ICCAD 2015 and includes eight large-scale circuits, containing up to more than one million standard cells. It is widely used for evaluating timing performance in both macro and global placement tasks. `ChiPBench` (Wang et al., 2024b) is a more recent and comprehensive benchmark suite for placement algorithms, which includes 20 circuits from diverse application domains, covering a broad range of design sizes and complexity levels. Detailed statistics for both suites are provided in Appendix D.1.

**Baselines**  As a general-purpose timing surrogate, LiTPlace can be integrated into existing gradient-based global placement (GP) frameworks as a plug-in objective term. We evaluate its effectiveness by incorporating it into three representative frameworks: **DREAMPlace** (Lin et al., 2019), **DREAMPlace 4.0** (Liao et al., 2022), and **Efficient-TDP** (Shi et al., 2025). DREAMPlace is one of the most widely used open-source GP frameworks, which accelerates placement via GPU-based gradient descent. DREAMPlace 4.0 is an updated version of DREAMPlace, which extends this framework by periodically invoking STA to reweight nets for timing optimization. Efficient-TDP represents a recent state-of-the-art (SOTA) timing-driven GP method that identifies and optimizes critical paths through periodic STA analysis.

Table 1: **Comparison of TNS ($\times 10^5$ps) and WNS ($\times 10^3$ps) for global placement derived by different approaches.** For both metrics, higher (closer to zero) is better. "+LiTPlace" represents integrating of our method into the baseline framework. For each comparison, the better results are highlighted in **bold red**. Improvement $= (S_{\text{Ours}} - S_{\text{Baseline}})/|S_{\text{Baseline}}|$, where $S$ is TNS or WNS.

| | DREAMPlace | | + LiTPlace | | DREAMPlace 4.0 | | + LiTPlace | | Efficient-TDP | | + LiTPlace | |
| | TNS | WNS | TNS | WNS | TNS | WNS | TNS | WNS | TNS | WNS | TNS | WNS |
|---|---|---|---|---|---|---|---|---|---|---|---|---|
| superblue1 | -262.44 | -18.87 | **-173.73** | **-16.88** | **-85.03** | **-14.10** | -95.44 | -15.05 | -17.44 | **-7.75** | **-12.56** | -7.93 |
| superblue3 | -76.64 | -27.65 | **-54.59** | **-26.80** | -54.74 | -16.43 | **-52.31** | **-14.12** | -20.40 | -11.82 | **-17.54** | **-11.02** |
| superblue4 | -290.88 | -22.04 | **-161.21** | **-18.89** | -144.38 | -12.78 | -144.88 | -13.39 | -82.88 | -9.17 | **-68.49** | **-6.94** |
| superblue5 | -157.82 | -48.92 | **-125.07** | **-38.78** | -95.78 | -26.76 | -98.38 | -25.97 | -62.18 | -24.65 | **-39.49** | **-22.91** |
| superblue7 | -141.55 | -19.75 | **-122.60** | **-17.17** | -63.86 | -15.22 | **-55.55** | **-15.22** | **-43.52** | **-15.22** | -49.53 | -15.22 |
| superblue10 | -731.94 | **-26.10** | **-687.58** | -28.71 | -768.75 | -31.88 | **-649.71** | **-25.13** | -558.14 | -23.08 | **-545.83** | **-22.53** |
| superblue16 | -453.57 | -17.71 | **-183.71** | **-14.10** | -124.18 | -12.11 | **-59.03** | -13.03 | -22.90 | **-8.63** | **-12.55** | -8.82 |
| superblue18 | -96.76 | -20.29 | **-64.81** | **-12.08** | -47.25 | -11.87 | **-42.99** | **-11.76** | -16.16 | -6.92 | **-13.92** | **-5.86** |
| Improvement | - | - | **30.0%** | **14.1%** | - | - | **9.7%** | **2.5%** | - | - | **17.9%** | **6.4%** |

**Evaluation Metrics**  In our experiments, we fix the positions of macros as provided in the benchmarks, and optimize the positions of cells. We report two standard timing metrics: **worst negative slack (WNS)** and **total negative slack (TNS)**, as introduced in Section 2.1. These metrics reflect the worst-case and cumulative timing violations, respectively, and are commonly used to assess timing performance. We use OpenTimer (Huang & Wong, 2015) for `ICCAD2015` and OpenSTA (OpenSTA, 2023) for `ChiPBench` to evaluate the timing performance.

**Training and Inference**  As `ICCAD2015` and `ChiPBench` have different technologies, we train a surrogate model for each benchmark suite. Each model is trained on a subset of circuits and evaluated on the full suite. For `ICCAD2015`, we use 4 circuits for training, 2 for validating, and 2 for zero-shot test. For `ChiPBench`, we use 12 circuits for training, 4 for validation, and 4 for zero-shot test. All splits are random. To construct training data, we generate three different placements per training/validation circuit using DREAMPlace, and use OpenSTA to extract delays. Each STA run provides delay labels for all edges in a circuit, so three placements per circuit correspond to only three STA executions while yielding hundreds of thousands to millions of labeled edges per design. The labeling cost is dominated by a small number of STA runs rather than by the total number of training samples. As a results, this produces approximately 1.2 million training samples for `ICCAD2015` and 3.2 million for `ChiPBench`. Each trained predictor is then tested on all circuits, including those unseen during training. Additional setup details are in Appendix D.2. The detailed model training procedure is provided in Appendix C.4. The training details—including training dataset and hyperparameters—are provided in Appendix D.2. The evaluation of the delay model, as part of the experimental analysis, is in Section 5.3.

## 5.2 MAIN RESULTS

Table 1 reports the TNS and WNS of different GP methods on `ICCAD2015`. The results show that LiTPlace consistently outperforms the baseline methods across most designs, with average improvements of 19.2% in TNS and 7.7% in WNS. More detailed results and the results on `ChiPBench` are in Appendix E.1. The visualizations of final placement outcomes are in Appendix E.2.

Table 2: Correlation coefficient.

**Correlation Analysis**  (1) Table 2 reports the prediction accuracy of trained models on the training set, unseen edges in trained layouts, unseen layouts of trained designs, and entirely unseen designs, where the accuracy is measured by the Pearson correlation coefficient between predicted values and ground-truth. The results show that

| | ICCAD2015 | ChiPBench |
|---|---|---|
| Training Set | 0.974 | 0.921 |
| Unseen Edges | 0.967 | 0.924 |
| Useen Layouts | 0.969 | 0.922 |
| Unseen Designs | 0.932 | 0.908 |

the model achieves high predictive accuracy and generalizes well even to unseen design instances. (2) Figure 4(a) presents the progression of the correlation coefficient during training, along with the corresponding TNS of the models at different training steps. The trends show that improving prediction accuracy indeed leads to better placement performance. (3) Figure 4(b) presents a correlation heatmap of metrics including TNS, WNS, HPWL, and the predicted total delay on selected $K$ critical paths, i.e., the additional timing term introduced in Equation (13). The results indicate that our timing term is more strongly correlated with TNS and WNS than HPWL. (4) These findings validate that the learned surrogate not only predicts delay accurately but also contributes directly to improving final placement quality. Additional training curves and detailed analyses are in Appendix E.3.

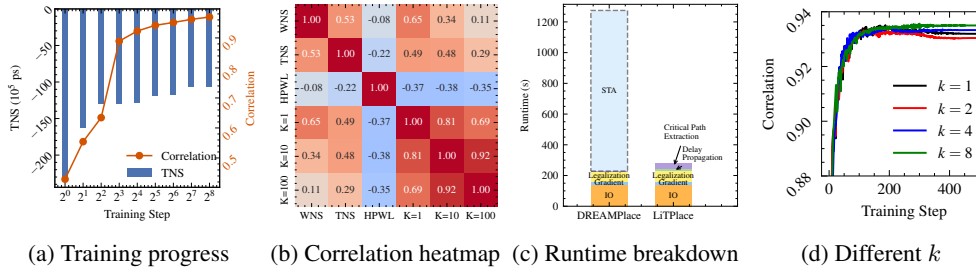

(a) Training progress    (b) Correlation heatmap    (c) Runtime breakdown    (d) Different $k$

Figure 4: Results of analytical experiments.

**Placement Runtime Breakdown** Figure 4(c) shows the placement runtime breakdown, indicating that the additional time introduces by our method, i.e., delay propagation and critical path extraction, is negligible, especially when compared with the extensive time required for STA.

### 5.3 ANALYSIS

Table 3: Improvement across splits.

|  | Training Set | Validation Set | Test Set |
|---|---|---|---|
| TNS | 23.80% | 12.40% | 17.00% |
| WNS | 6.90% | 7.80% | 9.10% |

**Ablation Study** As shown in Figure 4(d), setting the representation dimension as $k = 1$ is sufficient to achieve strong performance. This result is intuitive, because node and edge representations are primarily used to propagate timing-related information, and the key signal for delay computation is slew, which is indeed a scalar. We thus adopt $k = 1$ in main experiments for better efficiency. As shown in Table 3, LiTPlace achieves consistent improvement across all splits, indicating strong generalization beyond the circuits used for training. More ablation studies on different design choices are in Appendix E.4. Hyperparameter analysis results are in Appendix E.5. More discussions are in Appendix F.

## 6 CONCLUSION

This paper presents LiTPlace, a learning-based timing-driven global placement framework. At its core is a propagation-based GNN that serves as a differentiable surrogate model for predicting edge delays. The GNN architecture is carefully designed to enable efficient gradient computation with respect to cell coordinates, allowing seamless integration into gradient-based placement frameworks.

### ETHICS STATEMENT

This paper proposes a new algorithm for chip placement. We do not foresee any direct, immediate, or negative societal impacts of our research. We ensure that this work adheres to the ICLR Code of Ethics (https://iclr.cc/public/CodeOfEthics).

### REPRODUCIBILITY STATEMENT

All the results in this work are reproducible. We provide implementation details in Appendix C and experimental details in Appendix D to reproduce the results. Our code is available at https://anonymous.4open.science/r/LiTPlace-E201.

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

# A  SUPPLEMENTAL BACKGROUND

## A.1  PINS IN A CHIP

**Pins** serve as connection points for nets, facilitating signal transmission and communication across different design components. Pins can be broadly categorized into two types: internal pins and external pins.

**Internal pins** are those located on cells and macros within the chip's internal architecture. These pins are further divided into input pins and output pins. An **internal input pin** is a terminal through which a cell or macro receives signals from other components. An **internal output pin** is where a cell or macro sends out signals to other connected elements, propagating the results of internal computations or signal transformations.

**External pins** are positioned at the chip's boundaries, acting as interfaces between the chip and the external system environment. These pins can also be subdivided into external input pins and external output pins. An **external input pin** is where signals from external devices or circuits enter the chip. An **external output pin** is responsible for transmitting signals generated within the chip to external components.

## A.2  CLOCK SIGNAL

In most circuit designs, there is a dedicated clock pin used to receive an external clock signal. The clock signal is a periodic signal that serves as a timing reference for synchronizing the operation of different parts of the circuit. A clock signal consists of repetitive cycles, each with two key transitions known as **clock edges**. The **rising edge** occurs when the signal transitions from low to high. The **falling edge** happens when the signal transitions from high to low.

In a circuit, the primary components that receive the clock signal are registers. Specifically, when a register detects a specific clock edge, it captures the current input signal and produces a new output. Besides registers, some macros within the circuit also receive the clock signal, ensuring their operations are synchronized with the rest of the design.

## A.3  TIMING PATH

In a digital circuit design, a timing path is a specific trajectory that a signal travels through the circuit, governed by precise timing constraints to ensure proper operation. A timing path starts from a clearly identified point, which can be an external input pin, the clock pin of a register, or the output pin of a macro.

This starting point marks the initiation of the signal, either triggered by a clock edge or an external event. The signal then propagates through various combinatorial logic elements and interconnections, experiencing cell delays and net delays. These delays accumulate as the signal moves along the path, directly impacting its arrival time at the destination.

The timing path concludes at a specific endpoint, which can be an external output pin, the data input pin of a register, or the input pin of a macro. At this endpoint, the arrival time of the signal is measured against critical timing requirements, including setup time, hold time, etc.

Notably, the pins of a macro are considered the start or end points of timing paths because the macro is perceived as a "black box" from the perspective of external logic. Since its internal timing behavior is hidden, timing paths do not traverse through its internal logic, and timing analysis is limited to its pins.

## A.4  SLACK CALCULATION IN STA

In static timing analysis (STA), the calculation of slack is a critical step in evaluating the timing performance of a digital circuit.

The **arrival time** represents the amount of time required for a signal to propagate from the starting point of the timing path to a specific point in the circuit. This includes both cell delays and net delays. On the other hand, the **required arrival time** is the maximum allowable time for the signal to reach

its destination without violating timing constraints. This is derived from the clock period, setup or hold time requirements, and any additional constraints specified during design. It defines the upper limit for signal propagation to ensure proper synchronization and prevent timing violations.

The slack is then computed as the difference between the required arrival time and the arrival time. If the slack is positive or zero, the signal reaches its destination within the acceptable timing window, indicating that the path is timing-compliant. Conversely, a negative slack value indicates a timing violation, where the signal takes longer than the permissible duration to reach its endpoint, potentially causing functional errors.

## B RELATED WORK

### B.1 MACHINE LEARNING FOR ELECTRONIC DESIGN AUTOMATION (ML4EDA)

The use of machine learning (ML) in electronic design automation (EDA) has been extensively studied (MacMillen et al., 2000; Markov et al., 2012; Huang et al., 2021; Chen et al., 2024). ML techniques have been applied across various stages of the design flow, including RTL code generation and logic synthesis (Thakur et al., 2024; Wang et al., 2024a;c; Lai et al., 2025). Among them, ML-based methods for macro placement (MP) have attracted particular attention and are most relevant to our work.

Some approaches treat macro placement as a black-box optimization (BBO) problem and solve it using classical meta-heuristics such as simulated annealing (SA) and evolutionary algorithms (EA) (Kirkpatrick et al., 1983; Ho et al., 2004; Murata et al., 1995; Shi et al., 2023; Sherwani, 2012; Shunmugathammal et al., 2020; Vashisht et al., 2020; Murata et al., 1996; Chang et al., 2000; Roy et al., 2006; Khatkhate et al., 2004). Recently, reinforcement learning (RL) methods have emerged, beginning with AlphaChip (Mirhoseini et al., 2021), which dirst formulates macro placement as a Markov Decision Process (MDP). DeepPR (Cheng & Yan, 2021) and PRNet (Cheng et al., 2022) further integrate placement and routing, although they do not account for clock tree synthesis (CTS) or non-overlap constraints. MaskPlace (Lai et al., 2022) introduces the concept of wiremask, later extended by (Shi et al., 2023; Geng et al., 2024) to improve placement efficiency. ChiPFormer (Lai et al., 2023) applies offline RL to reduce online training cost. LaMPlace (Geng et al., 2025) extends this idea and proposes to learn a mask for optimizing cross-stage metrics. MaskRegulate (Xue et al., 2025) proposes to use RL as a regulator to guide timing optimization. These works focus on macro placement, where the number of objects is relatively small, making it more tractable for learning-based methods. In contrast, our work addresses timing-driven GP, significantly increasing problem scale and complexity, and pushing the boundary of ML4EDA toward finer-grained placement tasks.

### B.2 GLOBAL PLACEMENT

Most global placement algorithms adopt analytical methods, formulating objectives such as half-perimeter wirelength (HPWL) as differentiable functions of cell coordinates. These formulations are optimized via quadratic programming (Kahng et al., 2005; Viswanathan et al., 2007a;b; Spindler et al., 2008; Chen et al., 2008; Kim et al., 2012; Kim & Markov, 2012; Cheng et al., 2018) or direct gradient descent (Lin et al., 2019; 2020; Gu et al., 2020; Liao et al., 2022). Although highly efficient, these methods rely on heuristic proxies such as approximated HPWL, which——as shown in our experiments——may not correlate well with actual timing performance.

Since global cell distribution significantly impacts timing, timing-driven placement (TDP) extensions of analytical placers have been extensively studied, which can be broadly categorized into net-based and path-based approaches. **Net-based methods** modify net weights to guide placement toward better timing. Weight adjustment can be static or dynamically updated based on timing feedback (Burstein & Youssef, 1985; Chang et al., 2002; Dunlop et al., 1984; Eisenmann & Johannes, 1998; Obermeier & Johannes, 2004; Gao et al., 1992; Kahng et al., 2011; Luk, 1991). DREAMPlace 4.0 (Liao et al., 2022) uses a momentum-guided net-weighting scheme coupled with an on-the-fly timing engine to continuously steer placement toward timing improvement. **Path-based methods** explicitly extract timing paths and incorporate them into the optimization process, either as additional objective terms or constraints (Chowdhary et al., 2005; Jackson & Kuh, 1989; Swartz & Sechen, 1995). By maintaining an accurate path-level timing view, these methods can directly target critical-path delay reduction.

Guo & Lin (2022) propose Differentiable-TDP, a hand-crafted differentiable timing-driven framework, which approximates the STA process by manually designed differentiable proxies. Efficient-TDP (Shi et al., 2025) introduces a pin-to-pin attraction scheme that iteratively shortens distances between pins on high-slack nets, yielding substantial timing improvements with low integration overhead. However, these methods either rely on hand-craft heuristics or rely on running the STA process, which is time-consuming. To our knowledge, LiTPlace is the first to develop a learning-based, differentiable timing surrogate that can be efficiently integrated into gradient-based global placement.

Notably, Differentiable-TDP (Guo & Lin, 2022) has a similar motivation to our work. However, Differentiable-TDP still relies on hand-crafted analytical delay model. In contrast, our method is learning-based, avoiding delay-model-specific formulations and enabling adaptation to different delay models. Besides, our learning framework has the potential to be trained with post-routing timing data, which is a unique advantage. Moreover, since the differentiable STA engine in (Guo & Lin, 2022) requires full-graph propagation, it may require relatively high GPU memory demands. In contrast, we design our GNN architecture to maintain linearity, allowing us to compute gradients without backpropagating through the GNN.

## C  IMPLEMENTATION DETAILS

### C.1  GRAPH REPRESENTATION

We represent the circuit netlist as a directed acyclic graph (DAG), where each node corresponds to an input pin, and each directed edge represents a signal propagation path defined by timing dependencies. Specifically, the graph construction process is as follows:

**1. Standard Cell Modeling**  For standard logic cells, all *input pins* are represented as nodes in the graph. If the *output pin* $o_A$ of a cell $A$ is connected to an input pin $i_B$ of another cell $B$, a directed edge is established from the corresponding input pin $i_A$ of cell $A$ to the input pin $i_B$ of cell $B$.

In addition, we apply specific modeling strategies for special pins and boundary components to handle unique structural characteristics.

- **External Pin Modeling.** External input and output pins are also modeled as nodes. We add directed edges between these nodes and the connected internal cell pins according to their netlist connectivity.

- **Register Modeling.** For each register, we model its *clock pin (ck)* as a node in the graph. Clock signals originating from external input pins are ignored. The clock pin nodes of all registers are assigned a topological level of zero and starting points of signal propagation paths. If the *output pin* of a register connects to any standard cell input pin, we add a corresponding edge. The *input pin* of each register is also modeled as a node, serving as a terminal node with only ingoing edges but no outgoing edges. This is because input pins of registers are endpoints of timing paths.

- **Macro Modeling.** Each *pin* of a macro block is modeled as an independent node. However, no edges are created between pins within the same macro, as their internal timing structure is abstracted away at this level.

**2. Node Features**  Each node is represented by a 3-dimensional feature vector consisting of: (1) in-degree of the node, (2) out-degree of the node, and (3) the Capacitance of the corresponding pin (extracted from the `.lib` file).

**Edge Features**  Each edge is represented by a 5-dimensional feature vector derived from cell timing lookup tables (LUTs) via Principal Component Analysis (PCA).

- **Timing Lookup Table Structure.** In a given technology library (the `.lib` file), each edge is associated with four timing-related lookup tables describing the delay and slew characteristics for rising and falling transitions. Each table is a two-dimensional function of two indices (e.g., input slew and output load) and contains a grid of timing values.

- For ICCAD15 circuits, each lookup table table is a $7 \times 8$ table indexed by input slew and output load, with $7 + 8$ index numbers and $56$ entry values. There are four such tables, corresponding to rise/fall and delay/slew combinations, resulting in $L = 4 \times (7 + 8 + 7 \times 8)$ values for each edge.
- For ChiPBench circuits, each lookup table table is a $7 \times 7$ table indexed by input slew and output load, with $7 + 7$ index numbers and $49$ entry values. There are four such tables, corresponding to rise/fall and delay/slew combinations, resulting in $L = 4 \times (7 + 8 + 7 \times 8)$ values for each edge.

- **Feature Extraction.** We flatten the four timing tables of each cell into a single $L$-dimensional vector. PCA is then performed across all cell types in the library, and each cell is represented by a reduced 5-dimensional feature vector. These 5-dimensional vectors are used as the edge features in our netlist graph.

## C.2   3. MODEL ARCHITECTURE

The input of $\mathrm{MLP}_{\boldsymbol{\theta}}^{(1)}$ is the concatenation of $\boldsymbol{f}_u$, $\boldsymbol{f}_e$, and $\boldsymbol{f}_u^+$. The input dimensionality is therefore $3 + 5 + 3 = 11$. It outputs a vector with dimension $k^2 + 5k$. This output vector is then partitioned into three separate components with dimensions $k^2$, $4k$, and $k$, respectively. The first two components are further reMLPshaped into matrices of size $k \times k$ and $k \times 4$, which are used as the outputs $\boldsymbol{A}_e$ and $\boldsymbol{B}_e$, respectively. The third component is directly used as the output $\boldsymbol{c}_e$. Similarly, the input dimension of $\mathrm{MLP}_{\boldsymbol{\theta}}^{(2)}$ is also $11$, and its output is a vector of dimension $k + 5$. This vector is divided into three parts: a vector of dimension $k$, a vector of dimension $4$, and a scalar. These three parts are used as the outputs $\boldsymbol{\alpha}_e$, $\boldsymbol{\beta}_e$, and $\gamma_e$, respectively. $\mathrm{MLP}_{\boldsymbol{\theta}}^{(1)}$ and $\mathrm{MLP}_{\boldsymbol{\theta}}^{(2)}$ both consist of four hidden layers, each containing 32 neurons. Both $\mathrm{MLP}_{\boldsymbol{\theta}}^{(1)}$ and $\mathrm{MLP}_{\boldsymbol{\theta}}^{(2)}$ use the ReLU activation function for non-linear transformation.

We trained models with various $k$ settings. The results are shown in Figure 4(d) in the main text. Experimental results indicate that different values of $k$ have a little impact on the quality of the training outcomes. In fact, this conclusion is intuitive, as the main information that affect the timing metrics of downstream cells is the slew. Slew is a scalar, which is why $k = 1$ results in comparable results with larger $k$. To achieve a lightweight design, we select $k = 1$. Then, total number of learnable parameters in our model is $7,500$. Therefore, LiTPlace is very lightweight and parameter-efficient.

## C.3   PROOF OF THEOREM 1

Before we prove Theorem 1, we first prove the following lemma.

**Lemma 1.** *Given a circuit netlist $\mathcal{G} = (\mathcal{V}, \mathcal{E}, \mathcal{F})$, for any topological level $l \in \mathbb{N}$ and node $v \in \mathcal{V}^{(l)}$, there exists a set of matrices $\left\{ \boldsymbol{R}_{v,e'} \in \mathbb{R}^{k \times 4} : e' \in \bigcup_{i=0}^{l-1} \mathcal{E}^{(i)} \right\}$ and a vector bias $\boldsymbol{s}_v \in \mathbb{R}^k$, such that for any pairwise distance configuration $\boldsymbol{d}$, the node representation $\boldsymbol{h}_v^{(l)}$ satisfies:*

$$\boldsymbol{h}_v^{(l)} = \sum_{e' \in \bigcup_{i=0}^{l-1} \mathcal{E}^{(i)}} \boldsymbol{R}_{v,e'} \tilde{\boldsymbol{d}}_{e'}^+ + \boldsymbol{s}_v, \tag{14}$$

*where $\tilde{\boldsymbol{d}}_{e'}^+ \in \mathbb{R}^4$ is the pooled distance statistic vector associated with edge $e'$ (see Equation (3)).*

*Proof.* To prove this lemma, we employ **mathematical induction** on the topological level $l$.

We begin from $l = 0$. For a node $v_0 \in \mathcal{V}^{(0)}$, we have $\boldsymbol{h}_{v_0}^{(0)} = \boldsymbol{0}$. The conclusion holds naturally. We assume that the conclusion holds for topological level $l$, and we will show the conclusion for $l + 1$.

We consider a node $v \in \mathcal{V}^{(l+1)}$. For any edge $e = (u, v) \in \mathcal{E}$, it is trivial that $e \in \mathcal{E}^{(l)}$ and $u \in \mathcal{V}^{(l)}$. According to Equation (6) and the induction assumption, we have:

$$\begin{aligned}
\boldsymbol{h}_e^{(l)} &= \boldsymbol{A}_e \boldsymbol{h}_u^{(l)} + \boldsymbol{B}_e \tilde{\boldsymbol{d}}_e^+ + \boldsymbol{c}_e = \boldsymbol{A}_e \left( \sum_{e' \in \bigcup_{i=0}^{l-1} \mathcal{E}^{(i)}} \boldsymbol{R}_{u,e'} \tilde{\boldsymbol{d}}_{e'}^+ + \boldsymbol{s}_u \right) + \boldsymbol{B}_e \tilde{\boldsymbol{d}}_e^+ + \boldsymbol{c}_e \\
&= \sum_{e' \in \bigcup_{i=0}^{l-1} \mathcal{E}^{(i)}} \boldsymbol{A}_e \boldsymbol{R}_{u,e'} \tilde{\boldsymbol{d}}_{e'}^+ + \boldsymbol{A}_e \boldsymbol{s}_u + \boldsymbol{B}_e \tilde{\boldsymbol{d}}_e^+ + \boldsymbol{c}_e = \sum_{e' \in \bigcup_{i=0}^{l} \mathcal{E}^{(i)}} \boldsymbol{P}_{e,e'} \tilde{\boldsymbol{d}}_{e'}^+ + \boldsymbol{q}_e,
\end{aligned} \tag{15}$$

where

$$
\boldsymbol{P}_{e,e'} = \begin{cases} \boldsymbol{A}_e \boldsymbol{R}_{u,e'}, & e' \in \bigcup_{i=0}^{l-1} \mathcal{E}^{(i)}, \\ \boldsymbol{B}_e, & e' = e, \\ \boldsymbol{O}, & e' \in \mathcal{E}^{(l)} \setminus \{e\}, \end{cases} \quad \text{and} \quad \boldsymbol{q}_e = \boldsymbol{A}_e \boldsymbol{s}_u + \boldsymbol{c}_e. \tag{16}
$$

Then, according to Equation (8), we have

$$
\begin{aligned}
\boldsymbol{h}_v^{(l+1)} &= \frac{1}{|\mathcal{N}^-(v)|} \sum_{e=(u,v)\in\mathcal{E}} \boldsymbol{h}_e^{(l)} = \frac{1}{|\mathcal{N}^-(v)|} \sum_{e=(u,v)\in\mathcal{E}} \left( \sum_{e'\in\bigcup_{i=0}^{l}\mathcal{E}^{(i)}} \boldsymbol{P}_{e,e'} \tilde{\boldsymbol{d}}_{e'}^+ + \boldsymbol{q}_e \right) \\
&= \sum_{e'\in\bigcup_{i=0}^{l}\mathcal{E}^{(i)}} \left( \frac{1}{|\mathcal{N}^-(v)|} \sum_{e=(u,v)\in\mathcal{E}} \boldsymbol{P}_{e,e'} \right) \tilde{\boldsymbol{d}}_{e'}^+ + \frac{1}{|\mathcal{N}^-(v)|} \sum_{e=(u,v)\in\mathcal{E}} \boldsymbol{q}_e \\
&= \sum_{e'\in\bigcup_{i=0}^{l}\mathcal{E}^{(i)}} \boldsymbol{R}_{v,e'} \tilde{\boldsymbol{d}}_{e'}^+ + \boldsymbol{s}_v,
\end{aligned} \tag{17}
$$

where

$$
\begin{aligned}
\boldsymbol{R}_{v,e'} &= \frac{1}{|\mathcal{N}^-(v)|} \sum_{e=(u,v)\in\mathcal{E}} \boldsymbol{P}_{e,e'}, \quad \forall e' \in \bigcup_{i=0}^{l} \mathcal{E}^{(i)}, \quad \text{and} \\
\boldsymbol{s}_v &= \frac{1}{|\mathcal{N}^-(v)|} \sum_{e=(u,v)\in\mathcal{E}} \boldsymbol{q}_e.
\end{aligned} \tag{18}
$$

According to mathematical induction, the proof of Lemma 1 is completed.

$\square$

**Theorem 1.** *Given a circuit netlist* $\mathcal{G} = (\mathcal{V}, \mathcal{E}, \mathcal{F})$, *for any topological level* $l \in \mathbb{N}$ *and edge* $e = (u, v) \in \mathcal{E}^{(l)}$, *there exists a set of vectors* $\left\{ \boldsymbol{a}_{e,e'} \in \mathbb{R}^4 : e' \in \bigcup_{i=0}^{l} \mathcal{E}^{(i)} \right\}$ *and a scalar bias* $b_e \in \mathbb{R}$, *such that for any pairwise distance configuration* $\boldsymbol{d}$, *the predicted delay satisfies:*

$$
\hat{y}_e(\boldsymbol{d}) = \sum_{e'\in\bigcup_{i=0}^{l}\mathcal{E}^{(i)}} \boldsymbol{a}_{e,e'}^\top \tilde{\boldsymbol{d}}_{e'}^+ + b_e, \tag{11}
$$

*where* $\tilde{\boldsymbol{d}}_{e'}^+ \in \mathbb{R}^4$ *is the pooled distance statistic vector associated with edge* $e'$ *(see Equation (3)).*

*Proof.* According to Lemma (1), we can write

$$
\boldsymbol{h}_u^{(l)} = \sum_{e'\in\bigcup_{i=0}^{l-1}\mathcal{E}^{(i)}} \boldsymbol{R}_{u,e'} \tilde{\boldsymbol{d}}_{e'}^+ + \boldsymbol{s}_u \tag{19}
$$

for some $\boldsymbol{R}_{u,e'} \in \mathbb{R}^{k\times 4}$ and $\boldsymbol{s}_u \in \mathbb{R}^k$.

According to Equation (9), we have

$$
\hat{y}_e = \boldsymbol{\alpha}_e^\top \boldsymbol{h}_u^{(l)} + \boldsymbol{\beta}_e^\top \tilde{\boldsymbol{d}}_e^+ + \gamma_e = \boldsymbol{\alpha}_e^\top \left( \sum_{e'\in\bigcup_{i=0}^{l-1}\mathcal{E}^{(i)}} \boldsymbol{R}_{u,e'} \tilde{\boldsymbol{d}}_{e'}^+ + \boldsymbol{s}_u \right) + \boldsymbol{\beta}_e^\top \tilde{\boldsymbol{d}}_e^+ + \gamma_e \tag{20}
$$

$$
= \sum_{e'\in\bigcup_{i=0}^{l-1}\mathcal{E}^{(i)}} \left( \boldsymbol{R}_{u,e'}^\top \boldsymbol{\alpha}_e \right)^\top \tilde{\boldsymbol{d}}_{e'}^+ + \boldsymbol{\alpha}_e^\top \boldsymbol{s}_u + \boldsymbol{\beta}_e^\top \tilde{\boldsymbol{d}}_e^+ + \gamma_e \tag{21}
$$

$$
= \sum_{e'\in\bigcup_{i=0}^{l}\mathcal{E}^{(i)}} \boldsymbol{a}_{e,e'}^\top \tilde{\boldsymbol{d}}_{e'}^+ + b_e, \tag{22}
$$

where

$$
\boldsymbol{a}_{e,e'} =
\begin{cases}
\boldsymbol{R}_{u,e'}^{\top} \boldsymbol{\alpha}_e, & e' \in \bigcup_{i=0}^{l-1} \mathcal{E}^{(i)}, \\
\boldsymbol{\beta}_e, & e' = e, \\
\mathbf{0}, & e' \in \mathcal{E}^{(l)} \setminus \{e\},
\end{cases}
\quad \text{and} \quad b_e = \boldsymbol{\alpha}_e^T \boldsymbol{s}_u + \gamma_e. \tag{23}
$$

This completes the proof. $\qquad\square$

### C.4 TRAINING THE PREDICTOR

To train the predictor, we construct a dataset $\mathcal{D}$ using a collection of $C$ chip netlists $\{\mathcal{G}_c(\mathcal{V}_c, \mathcal{E}_c, \mathcal{F}_c)\}_{c=1}^{C}$. For each netlist, we generate a set of $M$ diverse layouts $\{\boldsymbol{X}_{c,m}\}_{m=1}^{M}$ using DREAMPlace. To avoid same layouts and keep diversity, we first run DREAMPlace to generate one layout, after which we randomly fix a subset of cells and then run DREAMPlace to complete the layout.

Next, we compute the delay corresponding to each edge using an EDA timing analysis tool, resulting in delay vectors $\boldsymbol{y}_{c,m} = \text{DelayCalc}(\mathcal{G}_c, \boldsymbol{X}_{c,m})$, where each entry represents the delay between the pair of input pins connected by the corresponding edge.

This yields the final dataset:

$$
\mathcal{D} = \{(\mathcal{G}_c, \boldsymbol{X}_{c,m}, \boldsymbol{y}_{c,m}) \mid c \in [C], \ m \in [M]\}. \tag{24}
$$

For each netlist $\mathcal{G}_c$ and layout $\boldsymbol{X}_{c,m}$, the predicted delay vector is given by

$$
\hat{\boldsymbol{y}}_{c,m} = \text{GNN}_{\boldsymbol{\theta}}(\mathcal{G}_c, \boldsymbol{d}(\mathcal{G}_c, \boldsymbol{X}_{c,m})), \tag{25}
$$

where $\boldsymbol{d}(\mathcal{G}_c, \boldsymbol{X}_{c,m})$ represents the vector of pairwise pins distances for all edges in $\mathcal{G}_c$ under layout $\boldsymbol{X}_{c,m}$. We use the MSE loss to train the predictor:

$$
\mathcal{L} = \frac{1}{CM} \sum_{c,m} \frac{1}{|\mathcal{E}_c|} \|\hat{\boldsymbol{y}}_{c,m} - \boldsymbol{y}_{c,m}\|^2, \tag{26}
$$

where $\|\cdot\|$ represents the Euclidean norm, i.e., the $\ell_2$ norm.

Notably, each edge in the netlist graph, under a specific layout, is assigned a label via EDA timing analysis tool. This layout-dependent, edge-level supervision offers fine-grained labels, resulting in high data efficiency per layout instance.

More experimental settings and details can be found in Appendix D.

### C.5 TIMING-AWARE GLOBAL PLACEMENT WITH TRAINED DELAY PREDICTOR

To effectively incorporate predicted timing information into the global placement process, we propose an efficient integration pipeline that leverages our trained delay predictor to guide cell placement toward timing-aware solutions. The core idea is to augment the traditional wirelength-based objective with a predicted total path delay term, updated periodically throughout placement.

Before placement begins, we precompute the following coefficients for each edge $e$, using the MLP component of our model conditioned on layout-independent features:

$$
\boldsymbol{A}_e \in \mathbb{R}^{k \times k}, \quad \boldsymbol{B}_e \in \mathbb{R}^{k \times 4}, \quad \boldsymbol{c}_e \in \mathbb{R}^k, \quad \boldsymbol{\alpha}_e \in \mathbb{R}^k, \quad \boldsymbol{\beta}_e \in \mathbb{R}^4, \quad \gamma_e \in \mathbb{R}.
$$

As these coefficients are independent of placement coordinates, they can be treated as constants once the netlist $\mathcal{G}$ and features $\mathcal{F}$ are fixed, and can be reused throughout the entire placement process.

As summarized in Algorithm 2, our integration procedure consists of three key stages——delay prediction, critical path extraction, and objective function integration——which are performed in every fixed number of steps during the placement flow. Below, we elaborate on each stage in detail.

**1. Delay Prediction and Path Extraction**  At specific placement iterations, we perform a full forward propagation of our trained delay model on the current layout to obtain the predicted delay for each edge in the graph. Thanks to the precomputed per-edge linear coefficients, this process only requires evaluating the forward propagation of a lightweight linear function, which is faster than full model inference.

Based on these delay values, we extract the top-$K$ timing-critical paths as follows:

- We traverse the DAG in topological order and compute the predicted arrival time for each node, defined as the maximum cumulative delay from any node at topological level 0 to the current node.
- We select the $K$ nodes with the largest predicted arrival times and backtrack from each to reconstruct the corresponding critical path.

**2. Linear Coefficient Precomputation**  Once the top-$K$ paths are identified, we compute the coefficients of the predicted total delay as a linear combination with bias. This step is performed via Algorithm 1, which operates in $\mathcal{O}(|\mathcal{E}|)$ time. Since the pre-edge linear coefficients have already been computed at initialization, the cost of this stage is minimal—especially when accelerated on GPU—and negligible compared to the critical path extraction process. We have included the runtime breakdown in the main text, demonstrating the high efficiency of our algorithm.

---

**Algorithm 1** Precomputation of Predicted Total Delay

---

**Require:** Netlist graph $G = (V, E)$, and for each edge $e \in E$, parameters: $A_e, B_e, c_e, \alpha_e, \beta_e, \gamma_e$
**Ensure:** The linear function (with a bias) $\boldsymbol{TotalDelay}$ of $d$
1:  $\boldsymbol{TotalDelay} \leftarrow 0$
2:  **for all** node $v \in V$ **do**
3:      Initialize $\boldsymbol{s}(v) \leftarrow \boldsymbol{0} \in \mathbb{R}^{1 \times k}$
4:  **end for**
5:  **for all** edges $e^{(l)} = \left(u^{(l)}, v^{(l+1)}\right) \in E$ in reverse topological order **do**
6:      $\boldsymbol{TotalDelay} \mathrel{+}= \boldsymbol{\beta}_{e^{(l)}}^{\top} \tilde{\boldsymbol{d}}_{e^{(l)}}^{+} + \gamma_{e^{(l)}}$
7:      $\boldsymbol{TotalDelay} \mathrel{+}= \left(\boldsymbol{s}\left(v^{(l+1)}\right)^{\top} + \boldsymbol{\alpha}_{e^{(l)}}\right)^{\top} \left(B_{e^{(l)}} \tilde{\boldsymbol{d}}_{e^{(l)}}^{+} + \boldsymbol{c}_{e^{(l)}}\right)$
8:      $\boldsymbol{s}\left(u^{(l)}\right) \mathrel{+}= \left(\boldsymbol{s}\left(v^{(l+1)}\right)^{\top} + \boldsymbol{\alpha}_{e^{(l)}}\right)^{\top} A_{e^{(l)}}$
9:  **end for**
10: **return** $\boldsymbol{TotalDelay}$

---

**3. Objective Function Integration**  The predicted total delay—expressed as a linear function—is incorporated into the placement objective over the following interval of iterations. To balance its influence with the wirelength objective, we adopt an adaptive gradient-based normalization strategy:

- Compute the $L_1$ norm of the gradient of the predicted delay term and the wirelength term.
- Their ratio is then multiplied by a user-defined hyperparameter $\eta$ to yield the final weighting coefficient for the delay term.
- This coefficient is recalculated every time when the critical paths are updated.

**4. Integration Pipeline**  The complete integration follows this pipeline:

- During the early placement stage, we perform regular placement iterations without incorporating timing.
- Starting from a specific iteration , and then at fixed intervals, we:
    1. Predict edge delays using our trained model.
    2. Extract top-$K$ critical paths using predicted arrival times.
    3. Precompute the delay term as a linear function.
    4. Integrate this term into the placement objective using the computed scaling coefficient.

- This process repeats at each interval, ensuring dynamic timing guidance during placement refinement.

This overall pipeline is detailed in Algorithm 2.

---

**Algorithm 2** LiTPlace Integration Pipeline

---

**Require:** Netlist graph $\mathcal{G} = (\mathcal{V}, \mathcal{E}, \mathcal{F})$, trained delay predictor $\text{GNN}_{\boldsymbol{\theta}}$, start iteration $T_0$, update interval $\Delta T$, max iterations $T_{\max}$, top-$K$ $K$, weight $\eta$
**Ensure:** Final placement $X$
 1: Initialize $X^{(0)}$
 2: Precompute per-edge linear coefficients: $\boldsymbol{A}_e, \boldsymbol{B}_e, \boldsymbol{c}_e, \boldsymbol{\alpha}_e, \boldsymbol{\beta}_e, \gamma_e$ for all $e \in E$ using $\text{GNN}_{\boldsymbol{\theta}}$
 3: **for** $t = 1$ **to** $T_{\max}$ **do**
 4:     $X \leftarrow X^{(t-1)}$
 5:     RunBaselinePlacementStep()
 6:     $\text{Obj} \leftarrow \text{WL}(X) + \lambda \text{D}(X)$
 7:     **if** $t > T_0$ **then**
 8:         **if** $(t - T_0) \bmod \Delta T = 0$ **then**
 9:             $\hat{y}_e \leftarrow \text{GNN}_{\boldsymbol{\theta}}(\mathcal{G}, X, \boldsymbol{A}_e, \boldsymbol{B}_e, \boldsymbol{c}_e, \boldsymbol{\alpha}_e, \boldsymbol{\beta}_e, \gamma_e)$         ▷ Predict edge delays
10:             $\mathcal{P}_K \leftarrow \text{ExtractPaths}(\hat{y}, K)$         ▷ Each path is a set of edges
11:             $\text{Delay}_{\text{total}} \leftarrow \text{PrecomputeDelay}(\mathcal{G}, \mathcal{P}_K)$         ▷ Build linear delay function
12:             $\mu \leftarrow \eta \dfrac{\|\nabla_X \text{WL}(X)\|_1}{\|\nabla_X \text{Delay}_{\text{total}}(X)\|_1}$
13:         **end if**
14:         $\text{Obj} \leftarrow \text{Obj} + \mu \text{Delay}_{\text{total}}(X)$
15:     **end if**
16:     $X^{(t)} \leftarrow \text{UpdatePlacement}(X, \text{Obj})$
17:     **if** $\text{Converged}(X^{(t)})$ **then**
18:         **break**
19:     **end if**
20: **end for**
21: **return** $X^{(t)}$

---

# D  EXPERIMENTAL DETAILS

## D.1  BENCHMARK STATISTICS

Table 4 and Table 5 detail the statistics for circuits from the `ICCAD2015` (Kim et al., 2015) and `ChiPBench` (Wang et al., 2024b) benchmark suites, respectively.

Table 4: Statistics of 8 circuits from the `ICCAD2015` benchmark suite.

| Circuit | #Macros | #Standard Cells | #Nets | #Pins |
|---------|---------|-----------------|-------|-------|
| superblue1 | 424 | 1215820 | 1215710 | 3767494 |
| superblue3 | 565 | 1219170 | 1224979 | 3905321 |
| superblue4 | 300 | 801968 | 802513 | 2497940 |
| superblue5 | 770 | 1090247 | 1100825 | 3246878 |
| superblue7 | 441 | 1937699 | 1933945 | 6372094 |
| superblue10 | 1629 | 984379 | 1898119 | 5560506 |
| superblue16 | 99 | 985909 | 999902 | 3013268 |
| superblue18 | 201 | 771845 | 771542 | 2559143 |

## D.2  EXPERIMENTAL SETUP

**Training Details** For the `ICCAD2015` dataset, we use the circuits `superblue1`, `superblue10`, `superblue16`, and `superblue18` for training; `superblue4` and

Table 5: Statistics of public benchmark circuits.

| Circuit | #Macros | #Standard Cells | #Nets | #Pins |
|---|---|---|---|---|
| ariane133 | 132 | 167907 | 197606 | 979135 |
| ariane136 | 136 | 171347 | 201428 | 1000876 |
| bp_fe | 11 | 33188 | 39512 | 185524 |
| bp_be | 10 | 51382 | 62228 | 293276 |
| bp | 24 | 307055 | 348278 | 1642427 |
| swerv_wrapper | 28 | 98039 | 113582 | 573688 |
| bp_multi | 26 | 152287 | 174170 | 813050 |
| vga_lcd | 62 | 127004 | 151946 | 706931 |
| dft68 | 68 | 41974 | 56217 | 226420 |
| or1200 | 36 | 26667 | 32740 | 153379 |
| mor1kx | 78 | 68291 | 81398 | 394210 |
| ethernet | 64 | 35172 | 44964 | 205739 |
| VeriGPU | 12 | 71082 | 85081 | 421857 |
| isa_npu | 15 | 427003 | 548451 | 2406579 |
| ariane81 | 81 | 153873 | 180516 | 894420 |
| bp_fe38 | 38 | 26859 | 32661 | 154162 |
| bp_be12 | 12 | 38393 | 47030 | 220938 |
| bp68 | 68 | 164039 | 191475 | 887046 |
| swerv_wrapper43 | 43 | 95455 | 110902 | 560088 |
| bp_multi57 | 57 | 127553 | 146710 | 680748 |

`superblue7` for validation; and hold out `superblue3` and `superblue5` as unseen test circuits. The partitioning is performed in a fully random manner.

For the `ChiPBench` dataset, we use `mor1kx`, `bp_be`, `swerv_wrapper`, `ariane81`, `or1200`, `bp68`, `bp`, `dft68`, `VeriGPU`, `swerv_wrapper43`, `ariane136`, and `bp_fe` for training; `bp_multi57`, `bp_multi`, `ethernet`, and `bp_be12` for validation; `ariane133`, `vga_lcd`, `isa_npu`, and `bp_fe38` as unseen test circuits. The partitioning is also fully random.

For each circuit, we generate three distinct placement layouts. The timing information for each layout is extracted using the OpenSTA tool. For each path, we select the maximum cell delay and net delay across process corners (e.g., rise/fall) to obtain a representative delay.

During training, we sample $100,000$ edges from each layout. For each design, two of its layouts are used for training and the third one is treated as an unseen layout of this design. Within each training layout, we split the sampled edges into $70\%$ for training and $30\%$ as unseen edges, which allows us to evaluate the model's generalization within the same layout.

Edge features are first reduced to five dimensions PCA, followed by z-score normalization using statistics computed from the training set. Node features are also normalized using statistics from the training set.

To ensure consistent label scaling across different circuits, we normalize edge delays by dividing them by the standard deviation of all delays across all layouts for each circuit. Additionally, edge distances are normalized by the die size, i.e., $(\text{die}_x + \text{die}_y)/2$.

To ensure numerical stability and facilitate learning of relative delay magnitudes, we introduce a trainable scaling factor for each circuit. This factor is multiplied with the predicted delay before computing the loss. The scaling factor is updated during training but is not used during inference. The loss function is the MSE between the scaled predicted delays and the ground-truth values.

We adopt cosine annealing to adjust the learning rate during training. For the `ICCAD2015` dataset, we use an initial learning rate of $5 \times 10^{-3}$ and a final learning rate of $1 \times 10^{-6}$ over 500 epochs. For the `ChiPBench` dataset, the initial and final learning rates are $5 \times 10^{-4}$ and $1 \times 10^{-6}$, respectively, also trained for 500 epochs.

All experiments are executed on a computational platform with an Intel Xeon Gold 6246R CPU (3.60 GHz) and NVIDIA RTX 3090 GPU. In the experiments on `ICCAD2015`, the training time was 2h17m34s; for `ChiPBench`, the training time was 2h39m54s.

**Integrated Global Placement Method Details**   As described in Section C.5, at each scheduled integration point, we apply our trained delay predictor to perform a forward pass and obtain per-edge delay estimates based on the current placement. Using these predictions, we extract the top-$K$ timing-critical paths and compute their total delay in linear form via Algorithm 1, which supports masked execution on arbitrary path subsets.

The resulting delay term is then incorporated into the placement objective for the next 15 optimization steps. Its influence is dynamically scaled using the gradient-based normalization strategy introduced in Section C.5: we compute the ratio between the $L_1$ norms of the wirelength and delay gradients, and multiply it by a user-defined coefficient $\eta$ to determine the final weighting.

In our experiments, we use one suite of hyperparameters across different designs. Specifically, we set the hyperparameters in Algorithm 2 as $T_0 = 400$, $\Delta T = 15$, $T_{\max} = 1000$, $K = 10000$, and $\eta = 0.1$.

# E  ADDITIONAL RESULTS

## E.1  SUPPLEMENTAL MAIN RESULTS

Table 6 presents the experimental results on `ChiPBench`. These results show similar conclusions with on `ICCAD2015`, demonstrating the effectiveness of LiTPlace across different datasets. Notice that previous timing-driven GP algorithms, such as DREAMPlace 4.0 and Efficient-TDP, fail to run on `ChiPBench` due to compatibility issues.

Table 7 presents the experimental results on `ICCAD2015`, including TNS, WNS, HPWL and runtime (RT). The results demonstrate that our method significantly improves the placement quality with only a slight increase in runtime.

Table 6: Complete Experimental Results on the `ChiPBench` Dataset. All units: TNS and WNS in ns, HPWL in $10^6$, and RT in seconds.

| Benchmark | DREAMPlace | | | | + LiTPlace | | | |
|---|---|---|---|---|---|---|---|---|
| | TNS | WNS | HPWL | RT | TNS | WNS | HPWL | RT |
| bp_be | -501.5 | -0.89 | 13.01 | 29.29 | **-445.1** | **-0.81** | 13.01 | 37.20 |
| bp_fe | -36.8 | -0.64 | 8.47 | 21.51 | **-29.2** | **-0.59** | 8.47 | 31.70 |
| dft_68 | -93.0 | **-0.55** | 9.08 | 29.87 | **-89.6** | **-0.55** | 9.08 | 37.52 |
| mor1kx | -11807.9 | -1.74 | 8.66 | 32.58 | **-11363.0** | **-1.70** | 8.72 | 41.51 |
| or1200 | -29607.6 | -45.46 | 4.48 | 22.32 | **-29301.8** | **-41.30** | 4.49 | 30.36 |
| swerv_wrapper | -19221.5 | -4.32 | 14.75 | 33.47 | **-17082.4** | **-2.73** | 14.75 | 43.22 |
| swerv_wrapper43 | -22758.2 | **-3.47** | 19.05 | 27.24 | **-19503.8** | -3.58 | 19.07 | 36.05 |
| ariane81 | -4666.2 | -3.50 | 24.68 | 32.54 | **-4086.0** | **-3.46** | 24.71 | 42.55 |
| ariane136 | -513954.0 | -35.73 | 25.49 | 34.26 | **-496675.0** | **-33.35** | 25.52 | 47.62 |
| bp | -42324.3 | -4.66 | 31.38 | 43.40 | **-37589.5** | **-4.60** | 31.64 | 65.00 |
| VeriGPU | -48886.8 | -14.55 | 5.41 | 32.60 | **-46117.7** | **-13.13** | 5.43 | 39.94 |
| bp68 | -25622.2 | -5.78 | 58.71 | 36.05 | **-17627.2** | **-4.67** | 58.93 | 57.90 |
| bp_fe38 | -5322.3 | -3.07 | 15.44 | 27.84 | **-4141.1** | **-2.73** | 15.46 | 28.52 |
| ariane133 | -5274.8 | -5.49 | 25.52 | 36.13 | **-4038.3** | **-5.10** | 25.76 | 48.31 |
| vga_lcd | -271064.0 | -5914.19 | 7.15 | 31.35 | **-259035.4** | **-5153.29** | 7.17 | 40.69 |
| bp_be12 | -674.0 | **-1.43** | 16.83 | 27.84 | **-662.6** | **-1.43** | 16.84 | 31.03 |
| bp_multi | -8407.8 | **-5.14** | 17.97 | 39.25 | **-8344.1** | **-5.14** | 17.97 | 56.82 |
| bp_multi57 | -2052.5 | -5.12 | 23.84 | 28.62 | **-1784.6** | **-4.59** | 23.92 | 45.40 |
| ethernet | -1.0 | -0.21 | 4.16 | 25.66 | **-0.9** | **-0.20** | 4.21 | 30.95 |
| isa_npu | -526.2 | -1.34 | 32.03 | 20.12 | **-400.0** | **-1.30** | 32.05 | 30.31 |
| Average Ratio | 1.14 | 1.09 | 1.00 | 0.76 | **1.00** | **1.00** | 1.00 | 1.00 |

Table 7: Complete experimental results on the `ICCAD15` dataset. TNS and WNS are evaluated with the common framework in (Guo & Lin, 2022) for fair comparison. Results of **Differentiable-TDP** (Guo & Lin, 2022), **Distribution-TDP** (Lin et al., 2024) and **TransPlace** (Hou et al., 2025) are from their original papers. Since **Distribution-TDP** and **TransPlace** does not report HPWL and runtime, those entries are left blank "-". The runtime of **Differentiable-TDP** is scaled from (Lin et al., 2024) to account for machine differences: $\text{RT} = \text{runtime\_from\_paper} \times \frac{\text{our\_DREAMPlace\_runtime}}{\text{DREAMPlace\_runtime\_from\_paper}}$.
The units: TNS in $10^5$ ps, WNS in $10^3$ ps, HPWL in $10^6$, and RT in seconds.

| Benchmark | Cadence Innovus | | | | TransPlace | | | |
|---|---|---|---|---|---|---|---|---|
| | TNS | WNS | HPWL | RT | TNS | WNS | HPWL | RT |
| superblue1 | -66.60 | -8.99 | 444.5 | 4655.00 | -255.76 | -20.1 | - | - |
| superblue3 | -92.50 | -20.36 | 477.5 | 4988.00 | -116.08 | -14.86 | - | - |
| superblue4 | -60.90 | -9.95 | 326.5 | 3035.00 | -303.17 | -16.41 | - | |
| superblue5 | -167.00 | -30.47 | 503.5 | 4397.00 | -129.57 | -25.17 | - | - |
| superblue7 | -140.00 | -13.24 | 605 | 9107.00 | -131.06 | -14.68 | - | - |
| superblue10 | -226.00 | -19.74 | 924.5 | 7100.00 | -1050.00 | -33.38 | - | - |
| superblue16 | -125.00 | -10.57 | 475.2 | 3832.00 | -211.45 | -13.65 | - | - |
| superblue18 | -87.10 | -7.64 | 245.7 | 3661.00 | -133.01 | -13.93 | - | - |
| Average Ratio | 4.39 | 1.25 | 1.03 | 4.54 | 8.21 | 1.71 | - | - |

| Benchmark | Differentiable-TDP | | | | Distribution-TDP | | | |
|---|---|---|---|---|---|---|---|---|
| | TNS | WNS | HPWL | RT | TNS | WNS | HPWL | RT |
| superblue1 | -74.85 | -10.77 | 432.8 | 596.20 | -42.10 | -9.26 | - | - |
| superblue3 | -39.43 | -12.37 | 478.4 | 837.12 | -26.59 | -12.19 | - | - |
| superblue4 | -82.92 | -8.49 | 312.2 | 361.84 | -123.28 | -8.86 | - | |
| superblue5 | -108.08 | -25.21 | 488.7 | 463.82 | -70.35 | -31.64 | - | - |
| superblue7 | -46.43 | -15.22 | 602.1 | 925.68 | -95.89 | -17.24 | - | - |
| superblue10 | -558.05 | -21.97 | 934.4 | 788.05 | -691.10 | -25.86 | - | - |
| superblue16 | -87.03 | -10.85 | 485.1 | 337.34 | -55.99 | -12.21 | - | - |
| superblue18 | -19.31 | -7.99 | 243.6 | 530.57 | -19.23 | -5.25 | - | - |
| Average Ratio | 2.80 | 1.17 | 1.02 | 0.55 | 2.19 | 1.19 | - | - |

| | DREAMPlace | | | | + LiTPlace | | | |
|---|---|---|---|---|---|---|---|---|
| | TNS | WNS | HPWL | RT | TNS | WNS | HPWL | RT |
| superblue1 | -262.44 | -18.87 | 422.0 | 176.61 | **-173.73** | **-16.88** | **410.7** | 208.45 |
| superblue3 | -76.64 | -27.65 | 478.2 | 229.05 | **-54.59** | **-26.80** | **458.7** | 278.94 |
| superblue4 | -290.88 | -22.04 | **312.0** | 120.82 | **-161.21** | **-18.89** | 322.5 | 166.37 |
| superblue5 | -157.82 | -48.92 | 488.3 | 208.76 | **-125.07** | **-38.78** | **476.0** | 245.31 |
| superblue7 | -141.55 | -19.75 | 604.3 | 257.82 | **-122.60** | **-17.17** | **591.1** | 316.55 |
| superblue10 | -731.94 | **-26.10** | 935.9 | 348.80 | **-687.58** | -28.71 | **908.0** | 414.22 |
| superblue16 | -453.57 | -17.71 | 435.8 | 98.56 | **-183.71** | **-14.10** | **421.8** | 131.89 |
| superblue18 | -96.76 | -20.29 | 243.0 | 93.11 | **-64.81** | **-12.08** | **234.1** | 135.95 |
| Average Ratio | 10.10 | 2.27 | 1.01 | 0.16 | **5.69** | **1.88** | **0.98** | 0.20 |

| | DREAMPlace 4.0 | | | | + LiTPlace | | | |
|---|---|---|---|---|---|---|---|---|
| | TNS | WNS | HPWL | RT | TNS | WNS | HPWL | RT |
| superblue1 | **-85.03** | **-14.10** | **443.1** | 1180.53 | -95.44 | -15.05 | 531.9 | 1278.73 |
| superblue3 | -54.74 | -16.43 | 482.4 | 1274.52 | **-52.31** | **-14.12** | **476.8** | 1313.48 |
| superblue4 | **-144.38** | **-12.78** | **335.9** | 1277.07 | -144.88 | -13.39 | 353.0 | 1316.68 |
| superblue5 | **-95.78** | -26.76 | 556.2 | 1251.72 | -98.38 | **-25.97** | **525.7** | 1300.73 |
| superblue7 | -63.86 | **-15.22** | 604.0 | 1399.17 | -55.55 | **-15.22** | 600.9 | 1460.51 |
| superblue10 | -768.75 | -31.88 | **1036.7** | 3040.44 | **-649.71** | **-25.13** | 1086.2 | 3101.36 |
| superblue16 | -124.18 | **-12.11** | **448.1** | 739.11 | **-60.69** | -13.03 | 460.2 | 794.26 |
| superblue18 | -47.25 | -11.87 | 253.6 | 597.64 | **-42.99** | **-11.76** | **246.2** | 636.19 |
| Average Ratio | 3.80 | 1.51 | **1.06** | 1.10 | **3.18** | **1.48** | 1.09 | 1.15 |

| | Efficient-TDP | | | | + LiTPlace | | | |
|---|---|---|---|---|---|---|---|---|
| | TNS | WNS | HPWL | RT | TNS | WNS | HPWL | RT |
| superblue1 | -17.44 | **-7.75** | 431.0 | 1062.47 | **-12.56** | -7.93 | **419.4** | 1265.22 |
| superblue3 | -20.40 | -11.82 | 472.5 | 1047.07 | **-17.54** | **-11.02** | **462.8** | 1135.35 |
| superblue4 | -82.88 | -9.17 | 326.8 | 1049.37 | **-68.49** | **-6.94** | **319.6** | 1365.61 |
| superblue5 | -62.18 | -24.65 | 520.2 | 1126.55 | **-39.49** | **-22.91** | **484.0** | 1174.33 |
| superblue7 | **-43.52** | **-15.22** | 600.8 | 1249.25 | -49.53 | **-15.22** | **597.4** | 1261.99 |
| superblue10 | -558.14 | -23.08 | 974.2 | 1876.40 | **-545.83** | **-22.53** | **912.3** | 1882.96 |
| superblue16 | -22.90 | **-8.63** | **459.9** | 665.20 | **-12.55** | -8.82 | 467.9 | 759.11 |
| superblue18 | -16.16 | -6.92 | 244.0 | 517.87 | **-13.92** | **-5.86** | **233.7** | 578.92 |
| Average Ratio | 1.28 | 1.09 | 1.03 | 0.91 | **1.00** | **1.00** | **1.00** | 1.00 |

## E.2 VISUALIZATION OF PLACEMENT

We provide the visualization the final placement outcomes of 28 chip designs from `ICCAD2015` and `ChiPBench` in Figure 5 and Figure 6, respectively.

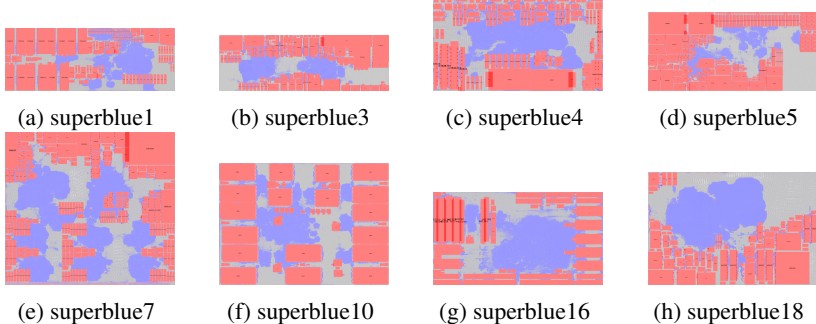

(a) superblue1    (b) superblue3    (c) superblue4    (d) superblue5

(e) superblue7    (f) superblue10    (g) superblue16    (h) superblue18

Figure 5: Visualization of final placement results of 8 designs from `ICCAD2015`.

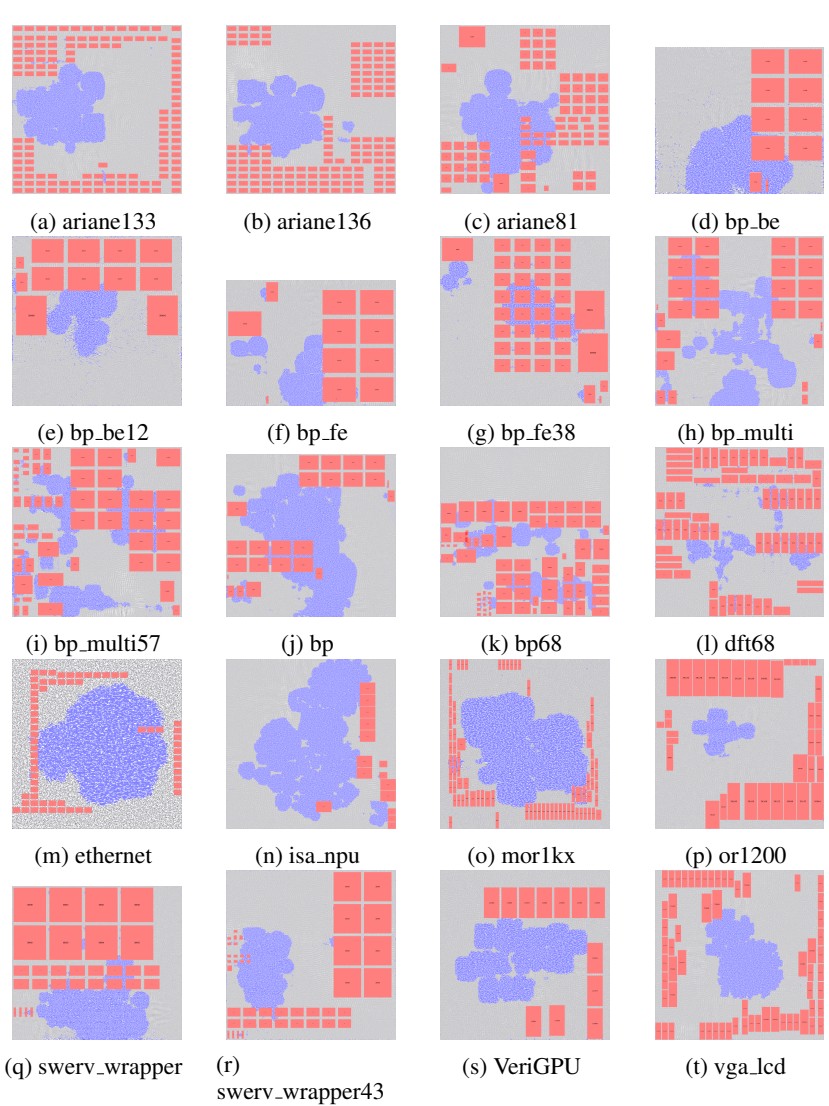

(a) ariane133    (b) ariane136    (c) ariane81    (d) bp_be

(e) bp_be12    (f) bp_fe    (g) bp_fe38    (h) bp_multi

(i) bp_multi57    (j) bp    (k) bp68    (l) dft68

(m) ethernet    (n) isa_npu    (o) mor1kx    (p) or1200

(q) swerv_wrapper    (r) swerv_wrapper43    (s) VeriGPU    (t) vga_lcd

Figure 6: Visualization of final placement results of 20 designs from `ICCAD2015`.

### E.3 CORRELATION ANALYSIS

In this section, we conduct correlation analysis experiments to show: (1) training improves prediction quality, (2) placement quality and prediction quality are improved together, (3) our learning-based optimization metric has a better correlation with the actual timing objectives than HPWL.

**Correlation Coefficient Improvement In Training Progress** We use the Pearson correlation coefficient between the predicted delay values and the groundtruth values to measure the prediction accuracy. Figure 7 and Figure 8 present the training curves of the correlation coefficient on `ICCAD2015` and `ChiPBench`, respectively.

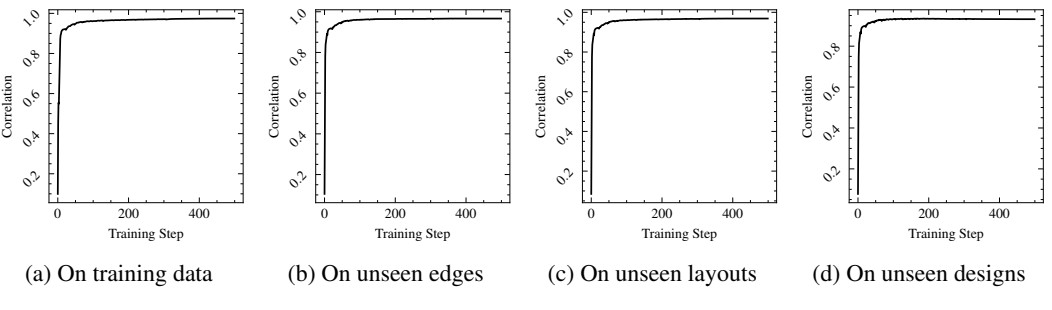

(a) On training data     (b) On unseen edges     (c) On unseen layouts     (d) On unseen designs

Figure 7: Training curve of `ICCAD` dataset

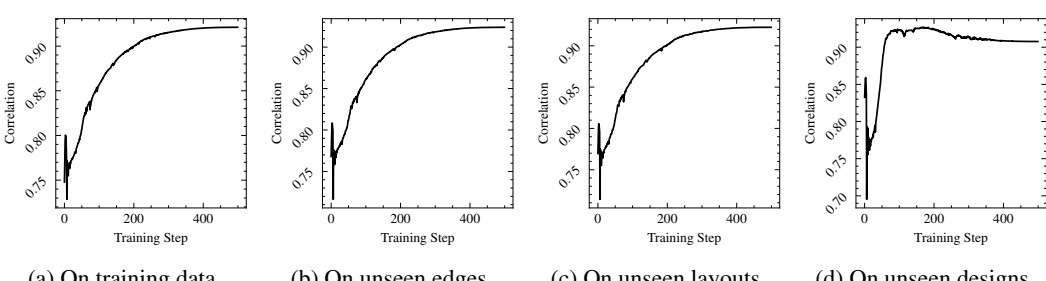

(a) On training data     (b) On unseen edges     (c) On unseen layouts     (d) On unseen designs

Figure 8: Training curve on `ChiPBench`.

**Correlation between Training Progress and Placement Quality** To investigate whether our training process indeed improves the final placement quality, we extract several model checkpoints during the training process and use them to perform timing-driven GP on the `superblue1` design. The TNS and WNS of the placement results, along with the correlation between the predicted delays on the validation circuits and the ground truth, are recorded in Figure 9. As shown in the figure, there is a clear trend indicating that better delay prediction generally leads to improved global placement results when our method is applied.

**Correlation Relationships Among Different Metrics** We investigate the pairwise correlation coefficients among the following metrics: (1) TNS and WNS, which are the actual optimization objectives, (2) HPWL, which is the most commonly used surrogate metric, (3) The total predicted delay of the top-$K$ critical paths, which is the additonal timing term used in our method, where $K = 1, 10, 100$. In our experiments, we collect these values for $48$ different placement solutions, derived from $6$ different methods and $8$ different designs in `ICCAD2015`. We use these datapoints to compute their pairwise correlation coefficients and plot a correlation heatmap, i.e., Figure 4(b) in the main text. Each number in the heatmap corresponds to the correlation coefficient between two metrics computed using the corresponding $48 \times 2$ datapoints. The results show that our proposed surrogate metrics, i.e., total delay of top-$K$ critical paths, exhibit a stronger correlation with WNS and TNS, compared to HPWL. This is why optimizing our additional term, rather than only optimizing HPWL, can effectively improve WNS and TNS.

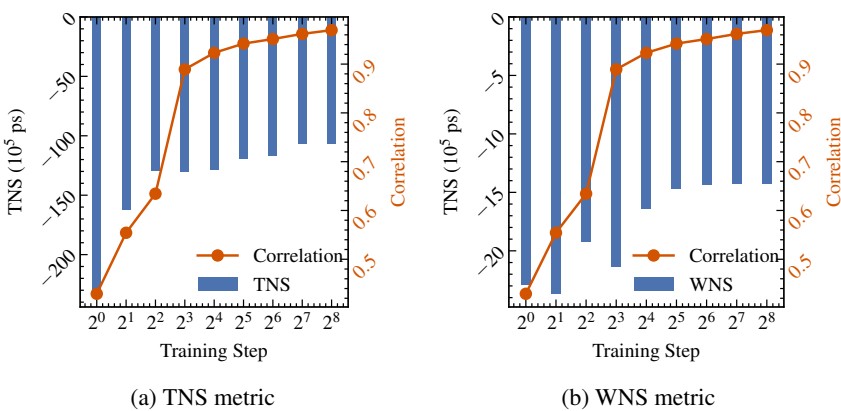

(a) TNS metric          (b) WNS metric

Figure 9: The superblue1 global placement results of different model checkpoints, along with the correlation coefficients between predictions and ground truth on validation circuits

Notably, in this correlation heatmap, HPWL almost does not correlate with WNS and TNS. This does not indicate that HPWL is totaly useless. This is because the placement datapoints that we collect are those with optimized HPWL values. The results show that when HPWL has been optimized to this level, further optimizing HPWL cannot improve TNS and WNS anymore. Instead, after we have obtained a solution with optimized HPWL, we should focus more on optimizing other timing-related metrics for further improvement.

### E.4  ABLATION STUDY

In this section, we conduct ablation studies to analyze the contributions of different design choices of the model.We train separate models under different settings to compare the resulting correlation coefficients.Subsequently, we evaluate their performance on real GP tasks by integrating each trained model into DREAMPlace + LiTPlace and testing on superblue1, superblue3, and superblue4:

1. $\ell_1$ loss: replace MSE with $\ell_1$ during training;

2. w/o capacitance: remove capacitance from node features;

3. w/o in-degree and out-degree: remove `in_deg`/`out_deg`;

4. w/o propagation: use a non-propagation variant that treats edges independently.

5. different $k$: use different representation dimension.

The training results are in Table 8 and the GP results are in Table 9 and Table 10. We also visualize the training curves w/ and w/o propagation.

Table 8: Correlation coefficient of the original model and the models under different setting.The results for different $k$ have already been presented in Figure 4(d).

|  | Training Set | Unseen Edges | Unseen Layouts | Unseen Designs |
|---|---|---|---|---|
| LiTPlace default model | 0.974 | 0.967 | 0.969 | 0.932 |
| $\ell_1$ loss | 0.958 | 0.953 | 0.957 | 0.924 |
| w/o capacitance | 0.936 | 0.931 | 0.934 | 0.904 |
| w/o in-degree and out-degree | 0.969 | 0.964 | 0.964 | 0.932 |
| w/o propagation | 0.956 | 0.949 | 0.949 | 0.916 |

As shown in Table 8 and Figure 10, the full model achieves the highest correlation across all splits, indicating stronger expressive capacity. Removing capacitance causes the largest degradation (e.g., $0.932 \rightarrow 0.904$ on *Unseen Designs*), highlighting its importance among node features. Eliminating in/out degree leads to a smaller drop and sometimes matches the full model on seen data. The non-propagation variant consistently underperforms, which is consistent with our design choice to model information flow via propagation.

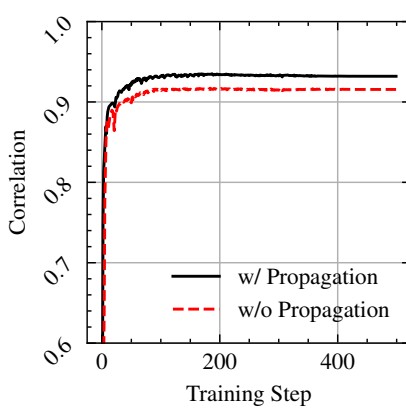

Figure 10: Training Curves for Models With and Without Propagation

Table 9 indicates that the original model is consistently the best or second best on TNS and WNS across all circuits. Table 10 further shows that placement quality is similar across $k$, while larger $k$ increases runtime; thus, $k = 1$ is a reasonable choice.

Table 9: Results of trained models under different settings applied to the GP task. The units: TNS in $10^5$ ps, WNS in $10^3$ ps, HPWL in $10^6$, and RT in seconds.

| Benchmark | original model | | $\ell_1$ loss | | w/o capacitance | | w/o in/out-degree | | w/o propagation | |
|---|---|---|---|---|---|---|---|---|---|---|
| | TNS | WNS | TNS | WNS | TNS | WNS | TNS | WNS | TNS | WNS |
| superblue1 | -173.73 | -16.88 | -188.54 | -16.76 | -262.34 | -18.64 | -262.34 | -18.64 | -275.36 | -21.83 |
| superblue3 | -54.59 | -26.80 | -56.33 | -27.56 | -61.25 | -31.32 | -61.25 | -31.32 | -59.20 | -32.99 |
| superblue4 | -161.21 | -18.89 | -167.54 | -19.11 | -171.36 | -19.21 | -167.64 | -19.20 | -180.21 | -19.06 |

Table 10: Results of trained models under different $k$ applied to the GP task. The units: TNS in $10^5$ ps, WNS in $10^3$ ps, HPWL in $10^6$, and RT in seconds.

| Benchmark | $k = 1$ | | | $k = 2$ | | | $k = 4$ | | | $k = 8$ | | |
|---|---|---|---|---|---|---|---|---|---|---|---|---|
| | TNS | WNS | RT | TNS | WNS | RT | TNS | WNS | RT | TNS | WNS | RT |
| superblue1 | -173.73 | -16.88 | 208.45 | -177.28 | -17.12 | 228.90 | -172.51 | -16.62 | 260.71 | -180.35 | -18.46 | 280.32 |
| superblue3 | -54.59 | -26.80 | 278.94 | -51.31 | -25.01 | 282.12 | -59.32 | -28.81 | 291.33 | -60.18 | -30.08 | 338.21 |
| superblue4 | -161.21 | -18.89 | 166.37 | -169.44 | -19.13 | 170.19 | -172.14 | -19.50 | 183.41 | -158.14 | -16.20 | 232.34 |

### E.5 INFLUENCE OF HYPERPARAMETERS IN GLOBAL PLACEMENT

We analyze the influence of hyperparameters on timing-aware global placement by conducting an ablation study on four key hyperparameters, as described in Section C.5:

- $K$: This parameter determines the number of top-k critical timing paths selected for modeling.
- $\Delta T$: This parameter determines the iteration interval for updating the critical path.
- Delay weight $\eta$: This coefficient controls the relative importance of the predicted total delay term in the placement objective. It helps adaptively determine the weight of our additional term, as described in Algorithm 2.
- `start_iter`: This parameter determines at which placement iteration the timing-aware objective is activated.

For $K$, we test different values of $K$ in the DREAMPlace + LiTPlace framework on the ICCAD2015. The experimental results are shown in the Table 11. These results show our method is robust to $K$. Empirically, setting $K = 10000$ yields both strong performance and efficient runtime.

For $\Delta T$, we adopt the same experimental setting as for $K$ and evaluate different values of $\Delta T$ within the DreamPlace+LiTPlace framework. The experimental results are reported in Table 12. The results show that as $\Delta T$ increases—that is, as the critical path set is updated less frequently—the runtime decreases while the timing quality degrades, which is consistent with intuition. Setting $\Delta T = 1$ strikes a good balance between runtime and timing performance. Using a smaller $\Delta T$ yields only marginal timing improvements while incurring additional runtime overhead.

Table 11: Ablation on the `Top-`$K$` parameter`using the `superblue` benchmarks. All units: TNS in $10^5$ ps, WNS in $10^3$ ps, HPWL in $10^6$, and RT in seconds.

| Benchmark | DREAMPlace | | | | $K=10$ | | | | $K=100$ | | | |
|---|---|---|---|---|---|---|---|---|---|---|---|---|
| | TNS | WNS | HPWL | RT | TNS | WNS | HPWL | RT | TNS | WNS | HPWL | RT |
| superblue1 | -262.44 | -18.87 | 422.0 | 176.61 | -298.91 | -24.27 | 410.7 | 180.33 | -240.57 | -22.06 | 410.4 | 187.25 |
| superblue3 | -76.64 | -27.65 | 478.2 | 229.05 | -64.44 | -25.05 | 457.2 | 234.87 | -62.44 | -25.60 | 457.2 | 241.54 |
| superblue4 | -290.88 | -22.04 | 312.0 | 120.82 | -191.14 | -21.61 | 312.3 | 128.7 | -186.96 | -19.99 | 311.8 | 136.37 |

| Benchmark | $K=1000$ | | | | $K=10000$ | | | | $K=100000$ | | | |
|---|---|---|---|---|---|---|---|---|---|---|---|---|
| | TNS | WNS | HPWL | RT | TNS | WNS | HPWL | RT | TNS | WNS | HPWL | RT |
| superblue1 | -263.25 | -18.95 | 410.6 | 192.72 | -173.73 | -16.88 | 410.7 | 208.45 | -237.30 | -18.73 | 410.8 | 318.98 |
| superblue3 | -61.27 | -25.06 | 457.2 | 254.53 | -54.59 | -26.80 | 458.7 | 278.94 | -47.34 | -25.66 | 458.7 | 465.15 |
| superblue4 | -175.07 | -19.28 | 312.1 | 143.28 | -161.21 | -18.89 | 312.0 | 166.37 | -159.87 | -19.23 | 312.2 | 277.08 |

Table 12: Ablation on the $\Delta T$ parameterusing the `superblue` benchmarks. All units: TNS in $10^5$ ps, WNS in $10^3$ ps, and RT in seconds.

| Benchmark | $\Delta T = 5$ | | | $\Delta T = 10$ | | | $\Delta T = 15$ | | | $\Delta T = 20$ | | |
|---|---|---|---|---|---|---|---|---|---|---|---|---|
| | TNS | WNS | RT | TNS | WNS | RT | TNS | WNS | RT | TNS | WNS | RT |
| superblue1 | -142.80 | -16.87 | 313.70 | -146.97 | -17.22 | 238.35 | -173.73 | -16.88 | 208.45 | -205.71 | -18.79 | 174.11 |
| superblue3 | -50.98 | -24.86 | 526.89 | -52.93 | -25.75 | 325.60 | -54.59 | -26.80 | 278.94 | -58.20 | -27.79 | 254.96 |
| superblue4 | -161.29 | -18.66 | 256.04 | -162.15 | -19.50 | 196.90 | -161.21 | -18.89 | 166.37 | -167.05 | -19.67 | 145.31 |

Then, we evaluate $\eta \in \{0.01, 0.05, 0.10, 0.50, 1.0\}$ and `start_iter` $\in \{0, 200, 400, 600\}$. We integrate LiTPlace into DREAMPlace and conduct experiments on the `superblue1` design. The baseline results from DREAMPlace are TNS $= -262.44$ and WNS $= -18.87$. The detailed results of our method under different hyperparameter configurations are reported in Table 14 (WNS) and Table 13 (TNS). The bold red numbers in the tables indicate the configurations that outperform the DREAMPlace baseline. The results demonstrate that our method consistently improves timing metrics across a wide range of hyperparameter settings, showing its robustness and practical effectiveness.

Table 13: TNS under different weights $\eta$ and `start_iter` values on superblue1

| weight $\eta$ | start_iter=0 | start_iter=200 | start_iter=400 | start_iter=600 |
|---|---|---|---|---|
| 0.01 | -266.06 | -262.83 | -261.23 | -282.48 |
| 0.05 | **-238.23** | **-238.34** | **-234.77** | **-255.34** |
| 0.10 | **-195.07** | **-179.51** | **-173.73** | **-167.89** |
| 0.50 | **-83.92** | **-87.56** | **-91.24** | **-92.62** |
| 1.00 | **-108.04** | **-131.71** | **-110.83** | **-155.71** |

# F  DISCUSSIONS

## F.1  COMPARISON WITH PRIOR ML-BASED PLACEMENT AND TIMING PREDICTION FRAMEWORKS

There has been several prior work on ML-based global placement and timing prediction. A natural question is whether these methods can be directly transferred or adapted to timing-driven global placement. This section provides a detailed comparison and clarifies why our framework is necessary rather than a modest extension of existing approaches.

Table 14: WNS under different weights $\eta$ and `start_iter` values on superblue1

| weight $\eta$ | start_iter=0 | start_iter=200 | start_iter=400 | start_iter=600 |
|---|---|---|---|---|
| 0.01 | -19.01 | -20.40 | -19.17 | -21.48 |
| 0.05 | -19.89 | -19.44 | **-18.69** | **-18.51** |
| 0.10 | **-17.66** | **-17.22** | **-16.88** | **-17.80** |
| 0.50 | **-14.26** | **-15.95** | **-17.18** | **-13.67** |
| 1.00 | **-14.11** | **-13.95** | **-15.46** | -30.33 |

### F.1.1 DIFFERENCES FROM ML-BASED CONGESTION-DRIVEN PLACEMENT METHODS

Prior work has investigated incorporating neural predictors into placement objectives in congestion-driven settings (Zheng et al., 2023; Liu et al., 2021; Hou et al., 2024). However, these methods cannot be adapted to timing-driven placement with only modest modifications, because the underlying problem formulation and modeling assumptions differ fundamentally. The main differences are summarized below.

**Congestion prediction is framed as spatial grid regression, whereas timing prediction is edge-level regression on a circuit graph.** Methods such as (Zheng et al., 2023; Liu et al., 2021) map geometric density features (e.g., RUDY, PinRUDY, MacroRegion) to a scalar field over 2D bins, i.e., $\mathbb{R}^{M \times N} \to \mathbb{R}^{M \times N}$. This image-like modeling is appropriate for spatially localized congestion. In contrast, timing estimation requires predicting per-edge delays in a DAG, where behavior depends on netlist topology rather than spatial proximity. These fundamentally different output spaces make congestion-driven predictors incompatible with timing-driven optimization.

**Timing exhibits long-range dependency along topologically ordered paths, whereas congestion is predominantly local.** Static timing analysis accumulates delays across long logic paths, often spanning tens to hundreds of gates. Models like RoutePlacer (Hou et al., 2024), which perform only two rounds of message passing, have small receptive fields that suffice for routability issues but cannot capture global timing dependencies.

**Scaling deep GNNs to million-node circuits is impractical for placement-loop inference.** Increasing message-passing depth to capture global dependencies incurs prohibitive compute and memory costs and also leads to over-smoothing Guo et al. (2022). Congestion predictors avoid these issues because their tasks rarely require deep propagation.

**LiTPlace uses STA-aligned propagation and enforces linearity in layout-dependent terms to enable analytic gradients.** Our propagation follows STA's topological order, naturally modeling long paths without deep GNN stacking. Further, the linear treatment of geometric inputs enables gradient computation without backpropagating through the entire GNN—an essential property for iterative placement. Such design requirements are absent in congestion-driven frameworks, making direct adaptation infeasible.

Therefore, timing-driven placement is not a minor variation of congestion-driven learning. The two differ in output space (edge vs. bin), dependency structure (global vs. local), scalability constraints (deep vs. shallow propagation), and their ability to integrate with gradient-based placement.

### F.1.2 DIFFERENCES FROM PRIOR NEURAL-NETWORK TIMING PREDICTORS

A number of recent works have proposed accurate and innovative timing predictors for various design stages, including TGNN (Guo et al., 2022), GNNTrans (Ye et al., 2023a), LSTP (Zheng et al., 2024), and EdgeGAT (Ye et al., 2023b). Although effective as standalone predictors, they are not suitable for integration into a gradient-based timing-driven placer. The core distinction is that LiTPlace is not merely a timing predictor, but a timing surrogate explicitly designed for optimization inside analytical placement.

**Existing timing predictors are not designed to serve as analytic placement objectives due to architectural complexity and expensive gradient computation.** TGNN, GNNTrans, LSTP, and EdgeGAT use deep GNN stacks, attention modules, or sequence models to capture global dependencies. While accurate, these models would require full backpropagation through many layers to obtain gradients with respect to cell coordinates—untenable for million-node circuits and repeated placement iterations.

**LiTPlace adopts STA-aligned propagation and enforces linearity in layout-dependent variables, eliminating the need for full GNN backpropagation.** As shown in Theorem 1, delay predictions are linear in the geometric inputs $\tilde{d}_e^+$, with coefficients determined solely by layout-independent cell and LUT features. This structure allows all coefficients to be precomputed once and enables analytic gradients during placement, dramatically reducing runtime and memory overhead compared with conventional timing predictors.

Thus, the difference lies not only in architecture but also in the optimization role the model plays. Existing timing predictors focus on maximizing inference accuracy for fixed layouts. LiTPlace, by contrast, is designed from the ground up to provide a differentiable timing term that can be jointly optimized with HPWL and density inside a timing-driven placer.

## F.2 BROADER IMPACT

Our proposed framework has the potential to substantially improve timing-driven global placement, a critical stage in physical design, by enabling learning-based delay modeling with analytical gradient support. In the context of modern semiconductor design, improving timing closure directly translates to fewer design iterations, shorter time-to-market, and enhanced energy efficiency. These benefits have broad economic implications, especially as process nodes shrink and timing margins tighten.

From a broader research perspective, our work bridges the fields of machine learning and electronic design automation (EDA), demonstrating how graph neural networks and learning-based methods can be used to approximate non-trivial circuit behaviors such as signal delay. This opens a new avenue for applying AI techniques to solve traditionally non-differentiable, domain-specific optimization problems at industrial scale.

Moreover, the modularity of our approach allows it to be integrated into existing placement flows with minimal modification. By serving as a plug-in objective for standard gradient-based placers, our method provides a practical path toward improving chip performance without redesigning the backend toolchain. This compatibility facilitates real-world adoption, making it a valuable stepping stone toward AI-augmented EDA pipelines.

## F.3 LIMITATIONS AND FUTURE DIRECTIONS

While LiTPlace demonstrates strong empirical results, several limitations remain and offer promising directions for future work.

**Support for Mixed-Size Placement.** Our current formulation focuses mainly on standard-cell global placement and does not yet support mixed-size scenarios involving both macros and standard cells. In industrial designs, macro placement significantly affects routing congestion and timing. Extending our framework to jointly handle mixed-size placement would expand its applicability and enable more holistic layout optimization.

**Beyond Timing Optimization.** At present, our objective focuses mainly on timing metrics such as WNS and TNS. In real-world deployments, additional factors such as power, routability, thermal reliability, and signal integrity are also important. Incorporating these multi-objective constraints into the learning and optimization process would further broaden the scope of AI-driven placement and move toward full-stack co-optimization.

**Toward End-to-End Differentiable EDA.** Our work represents a step toward building differentiable surrogates for traditionally non-differentiable components in the EDA flow. We envision future systems where multiple stages—placement, routing, buffering, and CTS—can be jointly optimized

through differentiable models. This direction may unlock new research paradigms where powerful foundation models are used to learn across entire chip design pipelines, tightly coupling algorithmic performance with downstream physical constraints.

### F.4 THE USE OF LARGE LANGUAGE MODELS

In accordance with ICLR 2026 policy, we disclose the use of Large Language Models (LLMs) as an assistive tool in the preparation of this manuscript. The primary application of LLMs was to aid in improving the clarity and quality of the writing.

Our process involved using an LLM to perform the following specific tasks:

- **Grammar and Spelling Correction:** Identifying and correcting grammatical errors and spelling mistakes.
- **Clarity and Readability Enhancement:** Rephrasing sentences and suggesting alternative phrasings to improve the overall readability and flow of the text.
- **Conciseness:** Assisting in shortening sentences and paragraphs to make the writing more direct and concise.

The core scientific contributions, analyses, and claims presented in this paper are the work of the human authors. We have ensured that the use of LLMs in the writing process was conducted responsibly and in line with academic and ethical standards.

