# OpenReview forum: "Learning A Linear Delay Surrogate Model for Timing-Driven Chip Global Placement"
_ICLR.cc/2026/Conference — Submitted to ICLR 2026_

### Official Review · Reviewer_Qsek · 2025-10-20

**Soundness:** 3
**Presentation:** 3
**Contribution:** 3
**Rating:** 6
**Confidence:** 2

**Summary:**

LiTPlace is proposed in this paper to predict delay with GNN, and is employed to serve as a differentiable surrogate integrated directly into gradient-driven global placement optimization. Experiments on publicly available chip design benchmarks demonstrate average improvements in both TNS and WNS, with negligible additional computational cost.

**Strengths:**

1. Proposes a differentiable delay prediction model that is directly integrated into gradient-based global placement, addressing a largely unexplored application of machine learning in timing-driven placement.

2. The proposed method achieves strong results when compared with GP solvers such as DREAMPlace and Efficient-TDP. The experimental results are convincing, and the paper is clearly written with a well-organized structure.

**Weaknesses:**

1. Due to the specialized nature of chip timing, the work may be difficult to fully understand for readers without a relevant background. Although the authors devote considerable effort to explaining the background, I personally still find chip timing concepts challenging to grasp. This is not a shortcoming of the authors.

2. LiTPlace is used within DREAMPlace to compute delays, but for the global placement problem, it is unclear whether there are other classical delay computation methods that could be used for comparison. The set of baseline methods appears limited.

**Questions:**

1. Since each model is trained on a subset of circuits and evaluated on the full benchmark suite, it would be helpful if the results explicitly indicate which chips were used for training, which for validation, and which for testing, along with the performance improvements observed for each set. Although the appendix provides this information, I believe it would be helpful to discuss the performance results for training, validation, and test sets directly in the main text.

2. The paper mentions that 'As ICCAD2015 and ChiPBench have different technologies, we train a surrogate model for each benchmark suite.' This raises questions about the generalization capability of the algorithm: for an unseen chip, can the current model be applied directly? It would be helpful to report the cross-benchmark performance, i.e., how a model trained on ICCAD2015 performs on ChiPBench, and vice versa.

---

> ### Author Response · Authors · 2025-11-20
> **Response to Reviewer to Qsek-- Part 1**
>
> Dear Reviewer Qsek,
>
> Thank you for your positive and insightful comments.. We sincerely hope our rebuttal could adequately address your concerns. If so, we would deeply appreciate it if you could consider raising your score. If not, please let us know your further concerns, and we will continue actively responding to your comments.
>
> **W1. Difficult to understand.**
>
> > Due to the specialized nature of chip timing, the work may be difficult to fully understand for readers without a relevant background. Although the authors devote considerable effort to explaining the background, I personally still find chip timing concepts challenging to grasp. This is not a shortcoming of the authors.
>
> Thank you very much for sharing this perspective. We understand that timing analysis is a highly specialized topic and may be unfamiliar to readers without prior EDA background. We have made our best efforts to provide necessary background explanations in **Section 2** and **Appendix A** to make the paper ccessible to readers with limited background, although the learning curve may still be steep. A comprehensive introduction to chip-level timing can be further found in [1].
>
> [1] Bhasker J, Chadha R. Static Timing Analysis for Nanometer Designs: A Practical Approach.
>
>
> **W2. Limited baseline methods.**
>
> > LiTPlace is used within DREAMPlace to compute delays, but for the global placement problem, it is unclear whether there are other classical delay computation methods that could be used for comparison. The set of baseline methods appears limited.
>
> Thank you for raising this question. We agree that evaluating timing-driven placement requires appropriate baselines that incorporate delay computation. We summarize below how we categorize existing timing-related approaches and why specific comparisons are made.
>
> - **Classical timing estimators exist, but many are designed for standalone prediction rather than integration into placement, making direct comparison inappropriate.** Prior neural timing models, such as [1,2] estimate timing for fixed layouts (pre-routing or post-routing) using deep GNN or attention-based architectures. These approaches aim for high-fidelity timing estimation rather than optimization. Using them inside global placement would require full backpropagation through deep networks at every iteration, imposing prohibitive overhead on million-node circuits. Thus, while valuable for prediction, they are not intended as alternative _delay computation engines_ for placement and are therefore not suitable baselines for our setting.
>
> - **LiTPlace is designed specifically for analytic timing optimization during placement, not just timing estimation.** As formalized in **Theorem 1**, geometric terms enter linearly while coefficients depend only on layout-independent features, allowing gradients to be computed without backpropagation through the entire model. This contrasts with classical predictors, which do not separate layout-dependent and independent components and therefore cannot efficiently serve as analytic placement objectives.
>
> - **Accordingly, our baselines focus on timing-aware placement frameworks rather than standalone timing predictors.** We compare against DREAMPlace4.0[3] and Efficient-TDP[4], which incorporate STA during placement to adjust objectives, and Differentiable-TDP[5], which embeds a differentiable analytical timing model. These are the closest existing approaches that _use timing information to guide placement_, making them the appropriate comparison class. In addition, following the suggestions of reviewers ur5J and BkgJ, we have also included the latest AI-based placement method TransPlace[6] as an additional baseline. LiTPlace achieves superior timing performance across benchmarks while maintaining compatibility with gradient-based optimization.
>
> [1] Guo et al., A timing engine inspired graph neural network model for pre-routing slack prediction. DAC 2022.
>
> [2] Zhong et al., Preroutgnn for timing prediction with order preserving partition: Global circuit pre-training, local delay learning and attentional cell modeling. AAAI 2024.
>
> [3] Liao et al., Dreamplace 4.0: Timing-driven global placement with momentum-based net weighting. DATE 2022.
>
> [4] Shi et al., Timing-driven global placement by efficient critical path extraction. DATE 2025.
>
> [5] Guo & Lin, Differentiable-timing-driven global placement. DAC 2022.
>
> [6] Hou et al., TransPlace: Transferable Circuit Global Placement via Graph Neural Network. KDD 2025.

---

> ### Author Response · Authors · 2025-11-20
> **Response to Reviewer to Qsek-- Part 2**
>
> **Q1. Dataset splitting and performance reporting.**
>
> > Since each model is trained on a subset of circuits and evaluated on the full benchmark suite, it would be helpful if the results explicitly indicate which chips were used for training, which for validation, and which for testing, along with the performance improvements observed for each set. Although the appendix provides this information, I believe it would be helpful to discuss the performance results for training, validation, and test sets directly in the main text.
>
> - Thank you for the helpful suggestion. For the ICCAD15 dataset, we split the circuits as follows:
>   - Training: superblue1, superblue10, superblue16, superblue18
>   - Validation: superblue4, superblue7
>   - Test: superblue3, superblue5
> This split was randomly determined.
>
> - The average timing improvements on the three subsets are summarized below:
>
>   | |Training set|Validation set|Test set|
>   |---|---|---|---|
>   |Average TNS improvement|23.80%|12.40%|17.00%|
>   |Average WNS improvement|6.90%|7.80%|9.10%|
>
>   The results show that LiTPlace achieves consistent improvement across all splits, indicating strong generalization beyond the circuits used for training.
>
> - We have incorporated this information into **Section 5.3** of the main paper to make dataset usage and generalization behavior more transparent.
>
>
>
> **Q2. Generalization capability.**
>
> > The paper mentions that 'As ICCAD2015 and ChiPBench have different technologies, we train a surrogate model for each benchmark suite.' This raises questions about the generalization capability of the algorithm: for an unseen chip, can the current model be applied directly? It would be helpful to report the cross-benchmark performance, i.e., how a model trained on ICCAD2015 performs on ChiPBench, and vice versa.
>
> Thank you for raising this important point regarding cross-technology generalization.
>
> - **The current model is not directly transferable across technologies because timing characteristics and library structures differ fundamentally between technology nodes.** In LiTPlace, edge-level timing features are extracted from cell timing Look-Up Tables (LUTs) in the `.lib` files using PCA. Different technologies provide LUTs with different dimensions, granularity, and electrical characteristics, meaning that a model trained on one technology does not receive compatible input features when applied to another. Generalizing across technology nodes would therefore require architectural adjustments rather than simple cross-evaluation.
>
> - **Cross-technology generalization is not the primary goal of this work, as timing-driven optimization is inherently technology-dependent and typically requires per-technology calibration in both research and industry.** In practice, commercial timing closure flows train or tune models separately for each technology because delay models, wire parasitics, and cell libraries change significantly across nodes. LiTPlace aligns with this deployment paradigm: our experiments demonstrate strong generalization across designs _within the same technology_, which is the relevant setting for timing-driven placement.
>
> - **Cross-node transfer remains possible with further extension but falls outside the current scope.** Achieving true cross-technology generalization may require technology-normalized representations or multi-domain training, which we consider promising future work for this paper.

---

> ### Author Response · Authors · 2025-11-27
> **We are looking forward to your feedback.**
>
> Dear Reviewer Qsek,
>
> We are writing as the authors of the paper "Learning A Linear Delay Surrogate Model for Timing-Driven Chip Global Placement" (ID: 15288). We sincerely thank you for your time and efforts during the rebuttal process. We are looking forward to your feedback to understand if our responses have adequately addressed your concerns. If so, we would deeply appreciate it if you could consider raising your score. If not, please let us know your further concerns, and we will continue actively responding to your comments. We sincerely thank you once more for your insightful comments and kind support.
>
> Best,
>
> Authors

---

### Official Review · Reviewer_BkgJ · 2025-10-27

**Soundness:** 3
**Presentation:** 3
**Contribution:** 2
**Rating:** 4
**Confidence:** 5

**Summary:**

This paper present This paper presents LiTPlace, a GNN-based framework aimed at improving timing-driven global placement (GP) in chip design. The paper proposes a new method using a differentiable GNN model to optimize timing in placement stage. LiTPlace predicts edge delays in chip layouts and integrates it as estimated timing objective directly into the optimization process. By integrating this GNN model in placement framework, it shows significant improvements in total negative slack (TNS) and worst negative slack (WNS), outperforming existing methods in both performance and computational efficiency.

**Strengths:**

The paper propose a differentiable method to directly optimize timing in placement without external tools such as OpenTimer to provide precise timing delay with STA.

LiTPlace has extensibility as it can be seamlessly integrated into modern placement framework such as DREAMPlace ,NTUPlace or ElfPlace.

**Weaknesses:**

There appears to be a lack of geometric-information modeling. Prior studies on circuit property prediction—e.g., LHNN [1] and CircuitGNN [2]—have shown that geometric information is crucial for accurate prediction. However, LiTPlace seems to omit geometric features. Is this because geometric structure is not important for timing prediction? Could the authors provide a detailed explanation?

Reference:

[1] Wang et al. LHNN: Lattice Hypergraph Neural Network for VLSI Congestion Prediction. DAC, 2022.

[2] Shu et al. Versatile Multi-stage Graph Neural Network for Circuit Representation. NeurIPS, 2022.

**Questions:**

Comparisons with prior work. Did the authors compare detailed runtime and performance between DTDGP [1] and LiTPlace? The paper reports ~50% improvement in WNS and ~70% in TNS, with ~1.8× speedup over a DREAMPlace 4.0–like method—please clarify the setup and metrics used. In addition, TransPlace [2] uses a GNN for transferable placement and can address congestion-aware and timing-aware objectives simultaneously. Could the authors provide a detailed comparison between TransPlace and LiTPlace? LiTPlace does not appear to be the first ML model targeting timing-driven placement.

Contributation of incorporating prediction into GP objective. Incorporating a neural network estimator to provide a differentiable objective and integrating it into the placement objective has been explored for congestion-driven settings [3,4,9]. What, specifically, distinguishes timing-driven placement from other objectives in the context of ML for placement? Can methods developed for congestion-driven placement be applied to timing-driven placement with modest changes (e.g., adding features or changing the prediction head)? If there are few or no such differences, why introduce a new framework instead of modestly adapting existing methods and applying them to timing-driven optimization?

Model design for timing prediction. Prior neural network-based timing-prediction work includes TGNN [5], GNNTrans [6], LSTP [7], and EdgeGAT [8]. How does the GNN used in LiTPlace differ from these models (e.g., architecture, input features, training targets, and generalization behavior)?

Reference:

[1] Guo et al. Differentiable-Timing-Driven Global Placement. DAC, 2022.

[2] Hou et al. TransPlace: Transferable Circuit Global Placement via Graph Neural Network. KDD, 2025.

[3] Zheng et al. Mitigating Distribution Shift for Congestion Optimization in Global Placement. DAC, 2023.

[4] Liu et al. Global Placement with Deep Learning-Enabled Explicit Routability Optimization. DATE, 2021.

[5] Guo et al. A Timing Engine Inspired Graph Neural Network Model for Pre-Routing Slack Prediction. DAC, 2022.

[6] Ye et al. Fast and Accurate Wire Timing Estimation Based on Graph Learning. DATE, 2023.

[7] Zheng et al. LSTP: A Logic Synthesis Timing Predictor. ASP-DAC, 2024.

[8] Ye et al. Graph-Learning-Driven Path-Based Timing Analysis Results Predictor from Graph-Based Timing Analysis. ASP-DAC, 2023.

[9] Hou et al. RoutePlacer: An End-to-End Routability-Aware Placer with Graph Neural Network. KDD, 2024.

---

> ### Author Response · Authors · 2025-11-20
> **Response to Reviewer to BkgJ -- Part 1**
>
> Dear Reviewer BkgJ,
>
> Thank you for your insightful and valuable comments. We sincerely hope our rebuttal could adequately address your concerns. If so, we would deeply appreciate it if you could consider raising your score. If not, please let us know your further concerns, and we will continue actively responding to your comments.
>
> **W1. Geometric information modeling.**
>
> > There appears to be a lack of geometric-information modeling. Prior studies on circuit property prediction—e.g., LHNN [1] and CircuitGNN [2]—have shown that geometric information is crucial for accurate prediction. However, LiTPlace seems to omit geometric features. Is this because geometric structure is not important for timing prediction? Could the authors provide a detailed explanation?
> >
> > Reference:
> >
> > [1] Wang et al. LHNN: Lattice Hypergraph Neural Network for VLSI Congestion Prediction. DAC, 2022.
> >
> > [2] Shu et al. Versatile Multi-stage Graph Neural Network for Circuit Representation. NeurIPS, 2022.
>
> - Thank you for your comments. **LiTPlace does incorporate geometric information through the distances statistics between pins**. These distances influence per-edge delay via $\tilde{d}_{e}^{+}$, which reflects geometric rela- **Layout-dependent features (geometric):** pairwise pin distance statistics extracted from the current layout;
>   - **Layout-independent features (non-geometric):** netlist topology, pin capacitance, node in/out-degrees, and standard-cell timing characteristics.tionships as cell positions change. As detailed in **Section 4.2**, the model takes two types of inputs:
>   - **Layout-independent features (non-geometric):** netlist topology, pin capacitance, node in/out-degrees, and standard-cell timing characteristics.
>
> - **We explicitly separate layout-dependent (geometric) and layout-independent (non-geometric) features to support efficient gradient-based placement.** Layout-independent properties remain fixed during placement, while geometric quantities vary dynamically as the layout evolves. Thus, delay is modeled to be linear with respect to geometric inputs, while the coefficients of this linear relationship are generated via nonlinear transformations of layout-independent timing features. Please refer to **Section 3** for more detailed motivations.
>
> - **This design enables analytic gradients and stable optimization rather than implying that geometric structure is unimportant.** Prior works such as LHNN and CircuitGNN focus on post-route congestion prediction or general-purpose circuit representation learning, where more complex geometric modeling is suitable. In contrast, LiTPlace is designed specifically for timing-driven _global placement_, where the surrogate must remain differentiable and computationally inexpensive at every optimization step.
>
>
> **Q1. Comparison with DTDGP.**
>
> > Did the authors compare detailed runtime and performance between DTDGP [1] and LiTPlace?
>
> - Thank you for your question. **We include DTDGP as a baseline and compare TNS, WNS, HPWL, and runtime results.** See the _Differentiable-TDP_ column in **Table 7, Appendix E.1**. Since DTDGP is not open-sourced, its TNS/WNS results are taken from the original paper, and runtime is estimated using: $RT = RT_{\text{paper}} \times
> \frac{RT_{\text{ours, DREAMPlace}}}{RT_{\text{paper, DREAMPlace}}}$. Note that Efficient-TDP + LiTPlace achieves overall much better timing performance than DTDGP.
>
> - **We also provide a detailed discussion in Appendix B.2 on the comparison with DTDGP.**
>   - DTDGP relies on a hand-crafted analytical delay model. In contrast, LiTPlace is learning-based, avoiding delay-model-specific formulations and enabling adaptation to different delay models. Besides, our learning framework has the potential to be trained with post-routing timing data, which is a unique advantage.
>   - Moreover, since the differentiable STA engine in DTDGP requires full-graph propagation, it may require relatively high GPU memory demands. In contrast, we design our GNN architecture to maintain linearity, allowing us to compute gradients without backpropagating through the GNN. However, as DTDGP is not open-sourced, we are unable to empirically verify memory usage, but this difference follows from the algorithmic formulation.

---

> ### Author Response · Authors · 2025-11-20
> **Response to Reviewer to BkgJ -- Part 2**
>
> **Q2. Setup and metrics.**
>
> > The paper reports ~50% improvement in WNS and ~70% in TNS, with ~1.8× speedup over a DREAMPlace 4.0–like method—please clarify the setup and metrics used.
>
> - Thanks for your question. As reported in our abstract and introduction, we achieve an average improvement of **19.2% in TNS** and **7.7% in WNS**. We did not claim the 50%, 70%, or $1.8\times$ numbers mentioned by the reviewer. Specifically, these values are obtained on the ICCAD15 dataset. According to the results in **Table 1**, our method brings an average improvement of 30.0% in TNS and 14.1% in WNS for DREAMPlace, 9.7% in TNS and 2.5% in WNS for DREAMPlace 4.0, and 17.9% in TNS and 6.4% in WNS for Efficient-TDP.
>
> - The detailed calculation method is as follows:
>   - For each of the above baselines, we measure the timing metrics (TNS, WNS) of each case before and after integrating LiTPlace, and compute the improvement brought by LiTPlace as $(S_{\text{Ours}} - S_{\text{Baseline}})/|S_{\text{Baseline}}|$, where $S$ denotes TNS or WNS.
>   - For each baseline, we then compute the average improvement of LiTPlace across all cases.
>   - Finally, we average the improvements across the three baselines to obtain the **19.2% (TNS)** and **7.7% (WNS)** values reported in the abstract.
>
> - We also report the runtime of each method (see **Table 7** in **Appendix E.1**). It is worth noting that although integrating LiTPlace introduces a small runtime overhead, it yields substantial improvements in timing performance.
>
> - Details regarding the setup and metrics used can be found in **Section 5.1**, and the complete experimental results---including those on the ChiPBench dataset---are provided in **Appendix E.1**.
>
> **Q3. Comparison with TransPlace.**
>
> > In addition, TransPlace [2] uses a GNN for transferable placement and can address congestion-aware and timing-aware objectives simultaneously. Could the authors provide a detailed comparison between TransPlace and LiTPlace? LiTPlace does not appear to be the first ML model targeting timing-driven placement.
>
> - Thank you for your valuable suggestion. **We have included TransPlace as an additional baseline.** We quote the results from the original TransPlace paper as follows:
>
>   |Benchmark|WNS|TNS|
>   |---|---|---|
>   |superblue1|-20.1|-25575.9|
>   |superblue3|-14.86|-11608.3|
>   |superblue4|-16.41|-30316.9|
>   |superblue5|-25.17|-12956.9|
>   |superblue7|-14.68|-13105.8|
>   |superblue10|-33.38|-105000|
>   |superblue16|-13.65|-21145|
>   |superblue18|-13.93|-13300.6|
>
>   These results are also reported in **Table 7 in Appendix E.1** of our paper.
>
> - **Notice that TransPlace addresses a fundamentally different task, as it learns a placement policy that directly predicts cell coordinates.** TransPlace formulates placement as a supervised prediction problem and outputs the final placement directly through a learned GNN model.
>
> - **LiTPlace does not predict placements; instead, it provides a differentiable surrogate for timing analysis that can be integrated into any gradient-based placer.** Our model approximates STA and supplies analytic timing gradients that allow existing analytical placers (e.g., DREAMPlace, Efficient-TDP) to explicitly optimize TNS/WNS. Thus, LiTPlace serves as an objective model rather than a placement generator.
>
> - **Because our method optimizes timing explicitly rather than producing placements heuristically, it achieves substantially stronger timing results.** When incorporated into DREAMPlace4.0 or Efficient-TDP, LiTPlace consistently achieves better TNS/WNS performance than the results reported by TransPlace across ICCAD2015 benchmarks.

---

> ### Author Response · Authors · 2025-11-20
> **Response to Reviewer to BkgJ -- Part 3**
>
> **Q4. Contributation of incorporating prediction into**
>
> > Incorporating a neural network estimator to provide a differentiable objective and integrating it into the placement objective has been explored for congestion-driven settings [3,4,9]. What, specifically, distinguishes timing-driven placement from other objectives in the context of ML for placement? Can methods developed for congestion-driven placement be applied to timing-driven placement with modest changes (e.g., adding features or changing the prediction head)? If there are few or no such differences, why introduce a new framework instead of modestly adapting existing methods and applying them to timing-driven optimization?
>
> We appreciate the reviewer’s question and have carefully examined prior ML-based placement methods, including the cited works. **These methods cannot be adapted to timing-driven placement with only modest modifications because the problem formulation and modeling requirements differ fundamentally.**
>
> - **Congestion prediction is framed as spatial grid regression, while timing prediction is edge-level regression on a circuit graph**. Methods such as [1,2] map geometric density features (e.g., RUDY, PinRUDY, MacroRegion) to a scalar field over placement bins, i.e., $\mathbb R^{M\times N}\rightarrow\mathbb R^{M\times N}$. This image-like formulation is suitable for congestion, which is spatially localized. In contrast, timing prediction requires estimating delays on edges in a directed acyclic graph, where behavior depends on netlist topology rather than spatial bins. Thus, the modeling assumptions of congestion predictors are not compatible with timing-driven optimization.
>
> - **Timing requires capturing long-range dependency across topologically ordered paths, whereas congestion is predominantly local.** STA accumulates delays across long logical paths, often spanning tens to hundreds of stages. In contrast, models like RoutePlacer [3] conduct only two rounds of message passing, yielding a small receptive field that suffices for local routability issues but cannot capture global timing dependencies.
>
> - **Scaling conventional deep GNNs to million-node circuits is impractical for placement loop inference.** Increasing message-passing depth to capture long-range dependencies leads to prohibitive compute and memory costs on large netlists, and suffers from over-smoothing [4]. Prior GNN-based congestion predictors do not address these scaling constraints because their tasks do not require deep propagation.
>
> - **LiTPlace adopts a propagation scheme aligned with STA, combined with a linear delay formulation to enable analytic gradients during placement.** Our propagation follows the STA topological order, providing natural long-path modeling without requiring deep GNN stacking. Moreover, our linear treatment of layout-dependent terms allows efficient gradient computation without backpropagation through the entire GNN—a crucial property for iterative placement. These design constraints are absent in congestion-driven frameworks, making direct adaptation insufficient.
>
> **Therefore, timing-driven placement is not a minor variation of congestion-driven learning, but a distinct problem requiring fundamentally different modeling assumptions.** The key differences lie in output space (edge vs. bin), dependency structure (global vs. local), computational requirements (deep vs. shallow propagation), and integration with gradient-based placement.
>
> [1] Zheng et al. Mitigating Distribution Shift for Congestion Optimization in Global Placement. DAC, 2023.
>
> [2] Liu et al. Global Placement with Deep Learning-Enabled Explicit Routability Optimization. DATE, 2021.
>
> [3] Hou et al. RoutePlacer: An End-to-End Routability-Aware Placer with Graph Neural Network. KDD, 2024.
>
> [4] Guo et al. A Timing Engine Inspired Graph Neural Network Model for Pre-Routing Slack Prediction. DAC, 2022.

---

> ### Author Response · Authors · 2025-11-20
> **Response to Reviewer to BkgJ -- Part 4**
>
> **Q5. Comparison with previous timing prediction works.**
>
> > Model design for timing prediction. Prior neural network-based timing-prediction work includes TGNN [5], GNNTrans [6], LSTP [7], and EdgeGAT [8]. How does the GNN used in LiTPlace differ from these models (e.g., architecture, input features, training targets, and generalization behavior)?
>
> Thank you for the question. **The key distinction is that LiTPlace is not merely a timing predictor, but a framework explicitly designed to be integrated into analytical, gradient-based placement.** Prior works focus on accurate timing estimation for fixed layouts at various design stages (e.g., logic synthesis, pre-routing, post-routing), whereas LiTPlace is constructed to serve as a low-overhead differentiable timing surrogate during iterative optimization.
>
> - **Existing timing predictors are not designed as analytic placement objectives due to architectural complexity and costly gradient computation.** TGNN, GNNTrans, LSTP, and EdgeGAT employ deep GNN stacks, attention mechanisms, or sequence models to capture global path dependencies. These architectures are effective as standalone predictors, but using them within a placement loop would require full backpropagation through many layers to obtain gradients w.r.t. cell locations, which is computationally expensive for million-node netlists and repeated iterations.
>
> - **LiTPlace adopts STA-aligned propagation and enforces linearity in layout-dependent variables to enable analytic gradients without full GNN backpropagation.** As shown in **Theorem 1**, the delay prediction for each edge is linear in geometry-related inputs $\tilde{d}_e^{+}$, while coefficients are computed from layout-independent timing features. This structure allows us to precompute coefficients once and compute gradients without full GNN backpropagation, significantly reducing both runtime and memory overhead during placement.
>
> **Thus, the difference lies not only in architecture, but in the optimization role the model plays.** Prior timing predictors aim to maximize inference accuracy for fixed layouts, whereas LiTPlace is designed from the outset to provide a differentiable timing term that can be jointly optimized with HPWL and density inside a timing-driven placer.

---

> ### Author Response · Authors · 2025-11-27
> **We are looking forward to your feedback.**
>
> Dear Reviewer BkgJ,
>
> We are writing as the authors of the paper "Learning A Linear Delay Surrogate Model for Timing-Driven Chip Global Placement" (ID: 15288). We sincerely thank you for your time and efforts during the rebuttal process. We are looking forward to your feedback to understand if our responses have adequately addressed your concerns. If so, we would deeply appreciate it if you could consider raising your score. If not, please let us know your further concerns, and we will continue actively responding to your comments. We sincerely thank you once more for your insightful comments and kind support.
>
> Best,
>
> Authors

---

### Official Review · Reviewer_ur5J · 2025-10-31

**Soundness:** 2
**Presentation:** 3
**Contribution:** 2
**Rating:** 2
**Confidence:** 3

**Summary:**

The paper proposes LiTPlace, a learning-based timing-driven global placement framework for VLSI design. LiTPlace introduces a propagation-based GNN that learns a differentiable linear delay surrogate to predict signal delays as an approximately linear function of geometric distances between connected cells. This surrogate enables direct gradient-based optimization of timing-aware placement. Experiments show average improvements of 19.2% in TNS and 7.7% in WNS over baseline frameworks.

**Strengths:**

1.	The idea of learning an approximate linear delay surrogate model for timing-driven global placement is novel.
2.	Demonstrates consistent improvement across ICCAD2015 and ChiPBench benchmarks and baseline GP approaches.
3.	The proposed framework is much more computing-efficient than DREAMPlace.

**Weaknesses:**

1. The proposed GNN model is a little bit confused. The methodology explains how the node and edge features are calculated at different typological level (different nodes and edges across the layout). But it does not provide a precise mathematical formulation of the message-passing and aggregation process for each GNN layer.
2. The propagation from one typological level to the next is a sequential linear transformation. I believe the entire model can be collapsed into one global linear propagation function. It does not exploit nonlinear relational reasoning or deep representation learning that typically justify a GNN. So, it is unclear whether the use of a GNN framework is necessary here.
3. As the delay prediction is the most important part to enable accurate timing-driven GP. I cannot find the details of the delay model training and evaluation, such as the training dataset size, training hyperparameter settings, and delay prediction accuracy.
4. While LiTPlace is compared against gradient-based EDA frameworks (DREAMPlace, Efficient-TDP), it ignores recent ML frameworks that also aim to learn placement objectives or surrogates such as TransPlace[1]
[1] Hou, Yunbo, et al. "TransPlace: Transferable Circuit Global Placement via Graph Neural Network." arXiv preprint arXiv:2501.05667 (2025).

**Questions:**

See weakness above.

---

> ### Author Response · Authors · 2025-11-20
> **Response to Reviewer ur5J --- Part 1**
>
> Dear Reviewer ur5J,
>
> Thank you for your insightful and valuable comments. We sincerely hope our rebuttal could adequately address your concerns. If so, we would deeply appreciate it if you could consider raising your score. If not, please let us know your further concerns, and we will continue actively responding to your comments.
>
> **W1.** **GNN** **message-passing and aggregation.**
>
> > The proposed GNN model is a little bit confused. The methodology explains how the node and edge features are calculated at different typological level (different nodes and edges across the layout). But it does not provide a precise mathematical formulation of the message-passing and aggregation process for each GNN layer.
>
> Thanks for your commands. We have provided the detailed description of our GNN in **Section 4.2**, and we further clarify here how it corresponds precisely to the standard message-passing framework.
>
> - **Message-passing GNNs are typically defined by two core functions---AGGREGATE and COMBINE---that update node representations through neighborhood information.** In classical formulations (Xu et al., 2019), each node first aggregates messages from neighbors
> $$
> m^{(l)}_v=\mathrm{AGGREGATE}(\\{h^{(l)}_u:u\in\mathcal N(v)\\}),
> $$
> and then combines this result with its previous state,
> $$
> h^{(l+1)}_v=\mathrm{COMBINE}(h^{(l)}_v,m^{(l)}_v).
> $$
> This aggregate-then-combine scheme defines the general message-passing paradigm.
>
> - **Our edge-based computation corresponds exactly to the message construction stage, with coefficients produced by an** **MLP** **from layout-independent timing features.** For each edge $e=(u,v)$, we compute
> $$
> h^{(l)}_e = A_e h^{(l)}_u + B_e \tilde d^{+}_e + c_e,
> $$
> ​where the coefficients $A_e,B_e,c_e$ are nonlinear transformations obtained from an MLP. This representation serves as the message passed from $u$ to $v$.
>
> - **Our aggregation step directly matches the** **AGGREGATE function** **in standard GNNs.** Each node collects messages from its incoming edges through
> $$
> m^{(l)}\_v = \frac{1}{|\mathcal N^{-}(v)|}\sum_{e=(u,v)} h^{(l)}_e,
> $$
> ​which is a standard neighborhood message-aggregation operator.
>
> - **Because our propagation follows the topological order of the DAG, the COMBINE function naturally reduces to assigning the aggregated message as the new node representation.** Under acyclic forward propagation, no additional transformation is needed, leading to $$ h^{(l+1)}_v = m^{(l)}_v, $$ consistent with simplified GNN variants where UPDATE is an identity map.
>
> - **Therefore, our model fully conforms to the message-passing GNN paradigm, with the only specialization being a linear treatment of layout-dependent terms to enable analytic gradients for placement.** Information is propagated along edges, aggregated at nodes, and updated layer-by-layer, while nonlinear MLP-generated coefficients encode timing-related circuit features. We have updated **Section 4.2** to make this correspondence explicit.
>
> [1] Xu et al., How powerful are graph neural networks? ICLR 2019.

---

> ### Author Response · Authors · 2025-11-20
> **Response to Reviewer ur5J --- Part 2**
>
> **W2. Why** **GNN** **is necessary.**
>
> > The propagation from one typological level to the next is a sequential linear transformation. I believe the entire model can be collapsed into one global linear propagation function. It does not exploit nonlinear relational reasoning or deep representation learning that typically justify a GNN. So, it is unclear whether the use of a GNN framework is necessary here.
>
> - Thank you for your insightful observations. Our model is linear only with respect to layout-dependent variables, but nonlinear with respect to layout-independent circuit features through MLP-generated coefficients, making the GNN essential.
> - Recall the following formulas in our method for node representations, edge representations, and predicted delays:
>
>   $ h_{v}^{(l+1)} = \sum_{e=(u, v)} \frac{h_{e}^{(l)}}{\mathcal{N}^{-}(v)} $
>
>   $ h_{e}^{(l)} = A h_{u}^{(l)} + B \tilde{d}_{e}^{+} + c $
>
>   $ \hat{y}\_e = \alpha^\top h_{u}^{(l)} + \beta^\top \tilde{d}_{e}^{+} + \gamma $
>
>   The terms involving geometric statistics $\tilde{d}_{e}^{+}$ are required to remain linear so that gradients with respect to placement coordinates can be computed analytically and efficiently. However, the coefficients that control this linear combination, $A_e$, $B_e$, $c_e$, $\alpha_e$, $\beta_e$, and $\gamma_e$, are not linear parameters. They are obtained from an MLP applied to layout-independent features of nodes and edges (e.g., LUT-based timing characteristics, pin capacitance, fan-out structure).
> - This design achieves two goals simultaneously:
>   - The MLP enables *nonlinear relational reasoning* over the circuit structure, capturing timing behavior that depends on the circuit features;
>   - The predicted delay is linear with respect to distance features, ensuring stable and efficient gradient computation with respect to coordinates for placement optimization.
> - We have updated **Section 4.2** to emphasize this separation of nonlinear circuit modeling and linear geometric modeling, and why this combination is necessary for timing-driven placement.
>
> **W3. Delay model training and evaluation details.**
>
> > As the delay prediction is the most important part to enable accurate timing-driven GP. I cannot find the details of the delay model training and evaluation, such as the training dataset size, training hyperparameter settings, and delay prediction accuracy.
>
> - Thank you for your kindful suggestion. Due to space limitations, we focus on the core idea, especially the task description and model architecture, in **Section 4**. The detailed model training procedure is provided in **Appendix C.4**. The training details---including training dataset and hyperparameters---are provided in **Section 5.1** and **Appendix D.2**. The evaluation of the delay model, as part of the experimental analysis, is in **Section 5.3**. Below we summarize the key information.
> - **Training Dataset Size (Section 5.1 & Appendix D.2)**
>   - ICCAD2015. We use superblue1, superblue10, superblue16, and superblue18 as training circuits.
>   - ChiPBench. We use 12 circuits, including mor1kx, bp_be, swerv_wrapper, ariane81, or1200, bp68, bp, dft68, VeriGPU, swerv_wrapper43, ariane136, and bp_fe.
>   - For both ICCAD2015 and ChiPBench, each dataset is randomly partitioned into training, validation, and test sets.
>   - During training, we sample 100,000 edges from each layout, and each design provides three layouts generated by DREAMPlace. We use two layouts for training, and the third one as an unseen layout to evaluate intra-design generalization. Within each training layout, the 100k edges are split into 70% training and 30% unseen edges, allowing us to evaluate generalization within the same layout.
> - **Training Hyperparameters (Appendix D.2)**
>   - We use an MSE loss to train the predictor (see **Appendix C.4** for the detailed formulation) and adopt a cosine annealing schedule for the learning rate.
>   - ICCAD2015: initial learning rate is $5 \times 10^{-3}$, final learning rate is $1 \times 10^{-6}$, and trained for 500 epochs.
>   - ChiPBench: initial learning rate is $5 \times 10^{-4}$, final learning rate is $1 \times 10^{-6}$, and trained for 500 epochs.
> - **Delay Prediction Accuracy (Section 5.3)**
>   - We evaluate prediction accuracy using the Pearson correlation coefficient across four scenarios: training edges, unseen edges in the same layout, unseen layouts, and fully unseen designs. As reported in **Table 2**, the correlation remains high (0.908–0.974), demonstrating strong generalization.
>   - **Figure 4(b)** further shows that our **predicted cumulative delay on top-K critical paths** is significantly more correlated with TNS/WNS than HPWL, highlighting the effectiveness of the learned surrogate in driving timing optimization.
> - We have updated the paper to improve cross-referencing so that these details are easier to locate.

---

> ### Author Response · Authors · 2025-11-20
> **Response to Reviewer ur5J --- Part 3**
>
> **W4. TransPlace baseline.**
>
> > While LiTPlace is compared against gradient-based EDA frameworks (DREAMPlace, Efficient-TDP), it ignores recent ML frameworks that also aim to learn placement objectives or surrogates such as TransPlace[1] [1] Hou, Yunbo, et al. "TransPlace: Transferable Circuit Global Placement via Graph Neural Network." arXiv preprint arXiv:2501.05667 (2025).
>
> - Thank you for your valuable suggestion. **We have included TransPlace as an additional baseline.** We quote the results from the original TransPlace paper as follows:
>
>    | Benchmark   | WNS    | TNS      |
>     | ----------- | ------ | -------- |
>     | superblue1  | -20.1  | -255.76  |
>     | superblue3  | -14.86 | -116.08  |
>     | superblue4  | -16.41 | -303.17  |
>     | superblue5  | -25.17 | -129.57  |
>     | superblue7  | -14.68 | -131.06  |
>     | superblue10 | -33.38 | -1050.00 |
>     | superblue16 | -13.65 | -211.45  |
>     | superblue18 | -13.93 | -133.01  |
>
>     These results are also reported in **Table 7 in Appendix E.1** of our paper.
>
> - **Notice that TransPlace addresses a fundamentally different task, as it learns a placement policy that directly predicts cell coordinates.** TransPlace formulates placement as a supervised prediction problem and outputs the final placement directly through a learned GNN model.
>
> - **LiTPlace does not predict placements; instead, it provides a differentiable surrogate for timing analysis that can be integrated into any gradient-based placer.** Our model approximates STA and supplies analytic timing gradients that allow existing analytical placers (e.g., DREAMPlace, Efficient-TDP) to explicitly optimize TNS/WNS. Thus, LiTPlace serves as an objective model rather than a placement generator.
>
> - **Because our method optimizes timing explicitly rather than producing placements heuristically, it achieves substantially stronger timing results.** When incorporated into DREAMPlace4.0 or Efficient-TDP, LiTPlace consistently achieves better TNS/WNS performance than the results reported by TransPlace across ICCAD2015 benchmarks.

---

> ### Author Response · Authors · 2025-11-27
> **We are looking forward to your feedback.**
>
> Dear Reviewer ur5J,
>
> We are writing as the authors of the paper "Learning A Linear Delay Surrogate Model for Timing-Driven Chip Global Placement" (ID: 15288). We sincerely thank you for your time and efforts during the rebuttal process. We are looking forward to your feedback to understand if our responses have adequately addressed your concerns. If so, we would deeply appreciate it if you could consider raising your score. If not, please let us know your further concerns, and we will continue actively responding to your comments. We sincerely thank you once more for your insightful comments and kind support.
>
> Best,
>
> Authors

---

### Official Review · Reviewer_CBAE · 2025-11-02

**Soundness:** 3
**Presentation:** 3
**Contribution:** 3
**Rating:** 6
**Confidence:** 3

**Summary:**

This paper introduces LiTPlace, a learning-based, differentiable timing-driven global placement framework for chip physical design.
The key idea is to train a propagation-based Graph Neural Network (GNN) that acts as a differentiable delay surrogate, enabling signal delays to be directly incorporated into gradient-based placement optimization.
By designing the GNN to have a linear propagation structure, the authors ensure that predicted delays are approximately linear functions of geometric distances, allowing analytical gradients and efficient optimization.
Evaluated on 28 benchmark designs (ICCAD2015 and ChiPBench), LiTPlace achieves 19.2% improvement in TNS and 7.7% improvement in WNS, with negligible wirelength increase.

**Strengths:**

1. The paper first introduces a differentiable timing-driven placement framework. Previous GP methods (e.g., DREAMPlace) optimized differentiable surrogates such as HPWL or density but could not directly optimize timing. LiTPlace is the first framework to embed timing objectives within a differentiable optimization loop through a learned delay model.

2. The proposed GNN simulates the signal propagation behavior of static timing analysis (STA). It follows topological levels and performs level-wise message passing, effectively capturing timing dependencies between upstream and downstream nodes.

3. Integrating LiTPlace into several baselines (DREAMPlace, DREAMPlace 4.0, Efficient-TDP) consistently improves timing metrics while adding minimal runtime overhead.

4. The paper introduces pooled distance statistics (min, max, mean) and aggregated successor features, allowing the model to reflect both geometric and electrical factors while remaining differentiable and computationally efficient.

**Weaknesses:**

1. While linearity improves efficiency, real-world timing behavior can exhibit strong nonlinearities (e.g., RC coupling, congestion effects). The model’s accuracy under such conditions remains uncertain, and no theoretical or empirical error bounds are provided.

2. The framework relies on a fixed number K of critical paths for optimization, but the selection and update frequency of K are treated as hyperparameters. This may affect optimization stability and lacks adaptivity across different designs.

3. Experiments only compare with open-source baselines and not with commercial-grade EDA tools (e.g., Synopsys ICC2, Cadence Innovus). Thus, the industrial applicability and scalability of LiTPlace remain unverified.

3. Training requires STA-generated delay labels for millions of samples. Despite reusing existing placements, this remains expensive in industrial design flows and may limit scalability.

4. The authors do not provide the code.

**Questions:**

1. In real designs, multiple signals can share the same net or pin, leading to coupling and congestion. Does LiTPlace handle these interactions, or is each signal path modeled independently?

2. What is the typical range of K, and how often is the critical path set updated? Have sensitivity analyses been conducted to evaluate its impact on convergence and final timing quality?

3. I am a new researcher working on AI4EDA placement, but I recently found that the ICCAD 2015 dataset link is no longer accessible. I would greatly appreciate it if you could share the dataset on an anonymous GitHub repository.

---

> ### Author Response · Authors · 2025-11-20
> **Response to Reviewer CBAE --- Part 1**
>
> Dear Reviewer CBAE,
>
> Thank you for your positive and insightful comments. We sincerely hope our rebuttal could adequately address your concerns. If so, we would deeply appreciate it if you could consider raising your score. If not, please let us know your further concerns, and we will continue actively responding to your comments.
>
> **W1. Linearity limits accuracy.**
>
> > While linearity improves efficiency, real-world timing behavior can exhibit strong nonlinearities (e.g., RC coupling, congestion effects). The model’s accuracy under such conditions remains uncertain, and no theoretical or empirical error bounds are provided.
>
> Thank you for raising this point. We clarify below how nonlinear timing effects are handled and why linearity applies only to the geometric component used for gradient computation.
>
> - **LiTPlace incorporates nonlinear behavior through MLP-generated coefficients rather than assuming a globally linear delay model.** The linear coefficients $A, B, c, \alpha, \beta, \gamma$ are produced by nonlinear neural networks based on layout-independent timing features such as pin capacitance, fan-in/fan-out, and standard-cell timing characteristics. Thus, nonlinear dependencies originating from cell libraries, logical depth, or intrinsic gate delays remain modeled in the parameters rather than the geometric term.
> - **Linearity is imposed only with respect to layout-dependent geometric features (specifically $\tilde{d}_e^{+}$), enabling analytic gradients while preserving expressiveness.** This structure allows gradients w.r.t. cell locations to be computed efficiently without backpropagating through the full GNN, which is critical for iterative timing-driven placement. While this introduces a modeling approximation, it replaces much coarser proxies such as HPWL, which ignores timing entirely. Our formulation thus strikes a balance between differentiability, scalability, and timing fidelity.
> - **Empirically, the learned timing surrogate correlates strongly with true timing metrics, despite nonlinear physical effects.** As shown in **Figure 4(b)**, the predicted timing term exhibits substantially higher correlation with TNS and WNS than HPWL. This suggests that the model captures meaningful timing behavior even under nonlinear conditions such as load variation and logic-path interactions.
>
> **W2. The choice of $K$.**
>
> > The framework relies on a fixed number K of critical paths for optimization, but the selection and update frequency of K are treated as hyperparameters. This may affect optimization stability and lacks adaptivity across different designs.
>
> Thank you for pointing this out. We examine the sensitivity of the framework to different values of $K$ below.
>
> - **We evaluate a range of $K$ values within the DREAMPlace + LiTPlace framework and report the results in Table 11.** Across all tested settings, the method consistently achieves strong timing improvement. Increasing $K$ generally yields slightly better timing quality at the cost of additional runtime, while smaller $K$ values reduce runtime with only modest degradation in timing performance. This provides a natural trade-off that users may select depending on optimization goals.
> - **We adopt $K=10,000$ in the main experiments because it offers a practical balance between runtime and timing quality.** As shown in **Table 11**, further increasing $K$ leads to diminishing improvements, indicating that the method is robust and does not require extensive tuning. In practice, simply setting $K=10,000$ works reliably across designs.
>
> Although $K$ is a hyperparameter, our results suggest that LiTPlace remains stable across a broad range of values, and the parameter encodes a meaningful runtime–quality trade-off rather than introducing instability.

---

> ### Author Response · Authors · 2025-11-20
> **Response to Reviewer CBAE --- Part 2**
>
> **W3. Comparison with commercial Tools.**
>
> > Experiments only compare with open-source baselines and not with commercial-grade EDA tools (e.g., Synopsys ICC2, Cadence Innovus). Thus, the industrial applicability and scalability of LiTPlace remain unverified.
>
> - Thank you for the thoughtful suggestion. We have added comparisons using the commercial placer Cadence Innovus, and summarize the results below:
>
>    |             | Innovus |          |       |      |
>     | ----------- | ------- | -------- | ----- | ---- |
>     |             | TNS     | WNS      | HPWL  | RT   |
>     | superblue1  | -66.6   | -8.9932  | 444.5 | 4655 |
>     | superblue3  | -92.5   | -20.3578 | 477.5 | 4988 |
>     | superblue4  | -60.9   | -9.9485  | 326.5 | 3035 |
>     | superblue5  | -167    | -30.4726 | 503.5 | 4397 |
>     | superblue7  | -140    | -13.244  | 605   | 9107 |
>     | superblue10 | -226    | -19.7424 | 924.5 | 7100 |
>     | superblue16 | -125    | -10.5669 | 475.2 | 3832 |
>     | superblue18 | -87.1   | -7.6445  | 245.7 | 3661 |
>
>    These results are also included in **Table 7, Appendix E.1** of the revised version.
>
> - Innovus achieves timing performance comparable to DREAMPlace4.0, but requires substantially longer runtime. **Efficient-TDP + LiTPlace achieves better overall timing results than Innovus while requiring significantly less runtime.** This demonstrates that LiTPlace not only maintains compatibility with analytical placers, but also delivers competitive timing quality relative to commercial tools. These new results strengthen the evidence that LiTPlace is scalable and effective in practical industrial settings.
>
> **W4. Required delay labels.**
>
> > Training requires STA-generated delay labels for millions of samples. Despite reusing existing placements, this remains expensive in industrial design flows and may limit scalability.
>
> Thank you for raising this concern. In our workflow, the labeling cost is in fact very low due to how samples are generated.
>
> - **A single** **STA** **run produces delay labels for all edges of the circuit simultaneously, and each edge serves as one training sample.** Our model does not require separate STA runs per sample. Instead, one STA execution per placement yields labels for millions of edges at once.
> - **Only a small number of placements are needed to obtain sufficient training data.** For example, on ICCAD2015, we train using four designs and generate three placements per design, leading to three STA runs per design. Across all designs, this produces roughly $1.2$ million training samples in total. We have clarified this detail in **Section 5.1**.
> - **Thus, the total labeling cost is dominated by a handful of STA runs rather than the number of samples.** This makes data collection practical even in industrial flows, where STA runs are routine and typically performed many times during timing closure.
>
> **W5. Code.**
>
> > The authors do not provide the code.
>
> Thank you for your suggestion. We have now released an anonymous repository containing the core code at the following link: [https://anonymous.4open.science/r/LiTPlace-E201](https://anonymous.4open.science/r/LiTPlace-E201). We will make it publicly available once the paper is accepted for publication.

---

> ### Author Response · Authors · 2025-11-20
> **Response to Reviewer CBAE --- Part 3**
>
> **Q1. Shared net or pin.**
>
> > In real designs, multiple signals can share the same net or pin, leading to coupling and congestion. Does LiTPlace handle these interactions, or is each signal path modeled independently?
>
> Thanks for your question.
> - **Our work follows the standard circuit model used in global placement literature, where each net has a single driver and multiple sinks, and each pin connects to exactly one net.** Under this commonly adopted assumption [1,2,3], logical nets do not contain multiple distinct signals, and pins are not shared across different nets. Thus, signal-level interaction does not arise from netlist connectivity itself.
> - **We understand that you might be** **asking** **whether** **LiTPlace accounts for interactions arising from routing congestion and** **coupling** **effects in the physical domain.** These effects do not correspond to multiple logical signals sharing the same net, but rather to competition for physical routing resources and parasitic effects between spatially adjacent wires.
>   - **Regarding congestion:** Similar to existing timing-driven placement methods, we enforce density regularization to prevent excessive clustering of cells, which reduces potential downstream congestion during routing.
>   - **Regarding** **coupling** **and other parasitic interactions:** Since our model is supervised using timing values produced by STA, any effects reflected in STA—including coupling, load variations, or net topology effects—may be learned by the prediction target and can be captured by the learned surrogate, even if not modeled explicitly through separate structural terms.
> - **Our current work trains on post-placement, pre-routing timing to balance fidelity and computational cost, but the framework can incorporate post-routing timing labels when available.** Training with post-route timing would allow the surrogate to learn more detailed physical effects (e.g., coupling and parasitic-induced delay), though at significantly higher labeling cost.
>
> [1] Lin et al., Dreamplace: Deep learning toolkit-enabled gpu acceleration for modern vlsi placement. DAC 2019.
>
> [2] Liao et al., Dreamplace 4.0: Timing-driven global placement with momentum-based net weighting. DATE 2022.
>
> [3] Shi et al., Timing-Driven Global Placement by Efficient Critical Path Extraction. DATE 2025.

---

> ### Author Response · Authors · 2025-11-20
> **Response to Reviewer CBAE --- Part 4**
>
> **Q2. K and critical path set updating.**
>
> > What is the typical range of K, and how often is the critical path set updated? Have sensitivity analyses been conducted to evaluate its impact on convergence and final timing quality?
>
> Thank you for the question. We clarify the choices of both hyperparameters and their influence on performance below.
>
> - In the main experiments, we set $K = 10000$ for all circuits, as this value provides stable performance while keeping path extraction overhead manageable. We include an ablation study on $K$, and the related discussions can be found in our response to Weakness 2.
>
> - In the main experiments, the critical path set is updated every $\Delta T = 15$ iterations. We further conduct an ablation study on $\Delta T$ as shown below:
>
>   - | $\Delta T$ | 5       |        |        | 10      |        |        | 15      |        |        | 20      |        |        |
>     | ---------- | ------- | ------ | ------ | ------- | ------ | ------ | ------- | ------ | ------ | ------- | ------ | ------ |
>     |            | TNS     | WNS    | RT     | TNS     | WNS    | RT     | TNS     | WNS    | RT     | TNS     | WNS    | RT     |
>     | superblue1 | -142.8  | -16.87 | 313.7  | -146.97 | -17.22 | 238.35 | -173.73 | -16.88 | 208.45 | -205.71 | -18.79 | 174.11 |
>     | superblue3 | -50.98  | -24.86 | 526.89 | -52.93  | -25.75 | 325.6  | -54.59  | -26.8  | 278.94 | -58.2   | -27.79 | 254.96 |
>     | superblue4 | -161.29 | -18.66 | 256.04 | -162.15 | -19.5  | 196.9  | -161.21 | -18.89 | 166.37 | -167.05 | -19.67 | 145.31 |
>
>   -  The results show that as $\Delta T$ increases---meaning the critical path set is refreshed less frequently---runtime decreases while timing quality degrades, which aligns with expectations. Setting $\Delta T = 15$ provides a good balance between runtime and timing performance.
>
> - **We observe that LiTPlace is** **robust** **to the choice of $K$ and $\Delta T$ within a reasonable range, and** **convergence** **behavior remains stable.** Similar hyperparameters appear in prior timing-driven placement frameworks such as Efficient-TDP, and our results indicate that LiTPlace performs consistently across broad settings without requiring fine tuning.
>
> - We have added these details, including ablation results and discussion, to **Appendix E.5** of the revised paper.
>
> **Q3. ICCAD 2015 dataset.**
>
> > I am a new researcher working on AI4EDA placement, but I recently found that the ICCAD 2015 dataset link is no longer accessible. I would greatly appreciate it if you could share the dataset on an anonymous GitHub repository.
>
> Thank you for pointing this out. We have also noticed that the original ICCAD 2015 download link is no longer accessible. The dataset can be downloaded from the following link:
>
> [https://drive.google.com/file/d/1xeauwLR9lOxnYvsK2JGPSY0INQh8VuE4/view?usp=sharing](https://drive.google.com/file/d/1xeauwLR9lOxnYvsK2JGPSY0INQh8VuE4/view?usp=sharing)
>
> This link was originally shared by the authors of Efficient-TDP [1]. We sincerely hope this helps.
>
> [1] Shi et al., Timing-Driven Global Placement by Efficient Critical Path Extraction. DATE 2025.

---

> ### Author Response · Authors · 2025-11-27
> **We are looking forward to your feedback.**
>
> Dear Reviewer CBAE,
>
> We are writing as the authors of the paper "Learning A Linear Delay Surrogate Model for Timing-Driven Chip Global Placement" (ID: 15288). We sincerely thank you for your time and efforts during the rebuttal process. We are looking forward to your feedback to understand if our responses have adequately addressed your concerns. If so, we would deeply appreciate it if you could consider raising your score. If not, please let us know your further concerns, and we will continue actively responding to your comments. We sincerely thank you once more for your insightful comments and kind support.
>
> Best,
>
> Authors

---

### Author Response · Authors · 2025-12-01
**Summary of Rebuttal -- Part 1**

Dear Area Chair,

We sincerely thank you for your careful work during the review and rebuttal process. We are also grateful to all reviewers for their constructive and insightful comments. While some reviewers assigned relatively low scores, in our assessment, no fatal flaws or fundamental limitations were identified. Most concerns focused on methodological clarification, experimental completeness, and additional comparisons. We have provided detailed responses to all questions and implemented all feasible suggestions. At this stage, we do not see any unresolved technical issues remaining.

For your convenience, we summarize below how we have addressed the main concerns raised by each reviewer.

**Response to Reviewer ur5J (Rating: 2)**

Reviewer **ur5J** acknowledged the novelty and efficiency of our linear surrogate formulation and raised questions regarding the GNN formulation and experimental details. We have addressed these points as follows:

- **Formalizationof the GNN methodology.** The reviewer was confused about the relationship between our GNN structure with the standard message-passing process (**W1**). We have explicitly aligned our formulation with the AGGREGATE and COMBINE operators for clarification.

- **Justification of architectural design**. The reviewer questioned the necessity of a GNN over a global linear function (**W2**). We explained why a GNN is necessary: the nonlinear circuit feature extraction is handled by MLPs, while geometric effects are propagated linearly for analytic gradient computation.

- **Trainingdetails.** Following the reviewer's suggestion (**W3**), we have added comprehensive specifications on dataset splits, hyperparameters, and confirmed high prediction accuracy.

- **Expanded comparative scope.** Following the reviewer's suggestion (**W4**), we have included a comparison with the recent GNN-based placement method TransPlace, demonstrating the superior performance of LiTPlace.


**Response to Reviewer BkgJ (Rating: 4)**

Reviewer **BkgJ** acknowledged the method's extensibility but requested deeper clarification on geometric modeling, distinctions from congestion-driven ML approaches, and broader comparisons. We have addressed these as follows:

- **Clarified geometric modeling.** The reviewer inquired about the explicit use of geometric features in our model (**W1**). We clarified that geometric information has been incorporated via distance statistics, which are treated linearly to support efficient gradient-based placement optimization.
- **Distinctionfrom prior works.** The reviewer asked for a clear distinction between timing-driven and congestion-driven ML approaches (**Q4**, **Q5**). We articulated fundamental differences in output space (edge-level vs. bin-level) and dependency structure (global vs. local), motivating the need for a specialized timing-driven framework.
- **Additionalbaselines.** Following the review's suggestion (**Q1**, **Q3**), we have provided detailed comparisons against additional baselines DTDGP and TransPlace.
- **Clarification of reported numbers.** Following the review's suggestion (**Q2**), we have clarified how performance improvements are computed.

---

> ### Author Response · Authors · 2025-12-01
> **Summary of Rebuttal -- Part 2**
>
> **Response to Reviewer CBAE (Rating: 6)**
>
> Reviewer **CBAE** positively assessed LiTPlace as a pioneering differentiable framework but emphasized the need for verification against industrial standards. We have addressed these as follows:
>
> - **Industrial applicability.** Following the reviewer's suggestion (**W3**), we have inclued the commercial tool **Cadence Innovus** as a baseline, demonstrating that LiTPlace can outperform the commercial tool.
>
> - **Handling of nonlinearity.** The reviewer asked how the linear assumption accounts for complex real-world timing behaviors (**W1**). We clarified that linearity applies only to geometric terms for differentiability, while physical complexities are captured by MLP-generated nonlinear coefficients.
>
> - **Hyperparameterrobustness.** Following the reviewer's suggestions (**W2, Q2**). We conducted sensitivity analyses on $K$ and update frequency $\Delta T$, confirming that the method remains stable across a wide range of settings.
>
> - **Labeling efficiency.** The reviewer raised concerns regarding the scalability of STA-based data generation (**W4**). We explained that a single STA run generates millions of edge labels simultaneously, ensuring the training cost is highly affordable even for industrial flows.
>
> - **Code and dataset.** Following the reviewer's suggestions (**W5, Q3**), we have released the anonymous code repository and provided the dataset link to facilitate reproduction.
>
>
> **Response to Reviewer Qsek (Rating: 6)**
>
> Reviewer **Qsek** found the results convincing but suggested improvements regarding baseline diversity, generalization analysis, and accessibility. We have addressed these as follows:
>
> - **Additional baselines.** The reviewer suggested expanding the set of baseline methods (**W2**). We expanded the evaluation to include **TransPlace** and **Cadence Innovus** , demonstrating that LiTPlace can outperform the commercial tool and other GNN-based method.
> - **Experimental setup.** The reviewer asked for explicit details on train/test splits to verify generalization (**Q1**). We listed the specific splits and reported consistent performance improvements across all subsets in the main text.
> - **Clarification of generalization scope.** The reviewer inquired about cross-technology generalization capabilities (**Q2**). We explained that cross-technology transfer requires retraining due to incompatible library definitions (e.g., LUT sizes), which is a standard constraint in physical design flows.
>
>
> We sincerely hope this summary is helpful and reduces your workload in assessing the rebuttals. Thank you again for your careful work during the review and rebuttal process.
>
> Best regards,
>
> Authors

---

### Meta-Review · Area_Chair_D8NF · 2026-01-04

**Summary:**

To address the challenge that time-series evaluation is highly complex and difficult to integrate into gradient-based generalization algorithms, this paper proposes LiTPlace. This framework learns a differentiable surrogate model to predict signal delays, thereby enabling time-aware gradient-based optimization. The four reviewers gave the following scores: 6, 6, 4, 2. Based on a comprehensive review and the author's rebuttal, my suggested decision is Reject.

**Reviewer Concerns:**

The following concerns have been addressed:
1. Industrial applicability. The authors used the commercial tool Cadence Innovus as a benchmark to demonstrate that LiTPlace outperforms this commercial tool.
2. The paper lacks precise mathematical formulas for the message passing and aggregation processes of the GNN model, and it also lacks detailed information on model training and evaluation, such as the size of the training dataset, training hyperparameter settings, and delay prediction accuracy. The author added relevant information in the rebuttal.
3. Comparison with more baseline models such as TransPlace. The author added relevant information in the rebuttal.

The following are some issues that may still exist:
The accuracy of the model remains uncertain, and no theoretical or empirical error range has been provided.
Adaptability of K selection and update frequency to different designs

**Reviewer Scores:**

Reviewer CBAE：6, this reviewer may not change the score.

Reviewer ur5J: 2, this reviewer may remain the same or be upgraded to 4.

Reviewer BkgJ：4, this reviewer is unlikely to raise the score due to his/her strong confidence.

Reviewer Qsek: 6, this reviewer may not change the score.

---

### Decision · Program_Chairs · 2026-01-26

Reject